# Piezo1 is an essential player in volume regulation of human glioblastoma cells

Maria Vittoria Leonardi, Angela Di Battista, Leonardo Donati 🄳, Luigi Sforna, Francesco Morena, Manlio Di Cristina 🄳, Luigi Catacuzzeno 🄳 and Antonio Michelucci 🄳

*Department of Chemistry, Biology, and Biotechnology, University of Perugia, Perugia, Italy*

Handling Editors: Peying Fong & Pawel Ferdek

The peer review history is available in the Supporting Information section of this article (https://doi.org/10.1113/JP289215#support-information-section).

**Abstract figure legend** Schematic summary of the proposed mechanism underlying Piezo1-mediated regulation of cell volume in U87-MG cells. Under isotonic conditions, activation of Piezo1 (i.e. by Yoda1) promotes the influx of $Na^+$ and $Ca^{2+}$, with $Na^+$ being the predominant ion. The resulting increase in intracellular $Na^+$ raises cytoplasmic osmolarity, driving osmotic water influx and leading to cell swelling. As cytosolic $Ca^{2+}$ accumulates, it activates both IK and BK channels. Concurrently, Piezo1-induced cell swelling also triggers the $Ca^{2+}$-independent activation of VRAC. The combined efflux of $K^+$ and $Cl^-$ via IK/BK channels and VRAC, respectively, promotes water loss and facilitates subsequent cell shrinkage.

M. V. Leonardi and A. Di Battista contributed equally to this work.

**Abstract** Cell volume regulation is a complex homeostatic process employed by nearly all cell types to adapt to osmotic perturbations and to support physiological processes such as proliferation and migration. This process is critically dependent on the activity of ion channels and transporters, which modulate intracellular water content through the controlled movement of osmolytes across the plasma membrane. We recently demonstrated in a human glioblastoma (GBM) cell line (U87-MG) that hypotonic-induced $Ca^{2+}$ influx through mechanosensitive channels is crucial for activating both large- (BK) and intermediate- (IK) conductance $Ca^{2+}$-activated $K^+$ ($K_{Ca}$) channels, which in turn drive the regulatory volume decrease (RVD) response. Although growing evidence implicates Piezo1 as the principal mechanosensitive channel for this $Ca^{2+}$ influx, definitive proof of its involvement in GBM cell volume regulation has been hindered by the lack of selective inhibitors. To address this issue, we generated a stable Piezo1 knock-out U87-MG cell line using the CRISPR-Cas9 technology. Our findings demonstrate that Piezo1 is essential for volume regulation during the hypotonic-induced RVD process. Specifically, Piezo1 ablation abolished the activation of the two predominant $K_{Ca}$ channels, IK and BK, in response to hypotonic stress, while leaving unalterd the function of the volume-regulated anion channel, which also contributes to RVD in these cells. Furthermore, pharmacological activation of Piezo1 with Yoda1 under isotonic conditions elicits a biphasic volume response: an initial swelling driven by $Na^+$ influx, followed by shrinkage mediated by $Ca^{2+}$-dependent activation of $K_{Ca}$ channels. Given the widespread expression of Piezo1 across animal cells, its role in volume regulation may represent a generalizable mechanism that extends beyond GBM cells.

(Received 10 May 2025; accepted after revision 22 July 2025; first published online 12 August 2025)

**Corresponding authors** A. Michelucci and L. Catacuzzeno: Department of Chemistry, Biology, and Biotechnology, University of Perugia, Perugia, Italy. Email: antonio.michelucci@unipg.it, luigi.catacuzzeno@unipg.it

## Key points

- Piezo1 is essential for cell volume regulation in human GBM U87-MG cells, particularly during RVD following hypotonic swelling.
- Piezo1 mediates RVD via $Ca^{2+}$-dependent activation of IK and BK channels. Knock-out of Piezo1 abolishes hypotonic-induced $K_{Ca}$ channel activation without altering their expression, indicating that Piezo1-driven $Ca^{2+}$ influx is a key upstream signal.
- Under isotonic conditions, Piezo1 activation induces volume a biphasic volume response: initial swelling via $Na^+$ influx, followed by $Ca^{2+}$-mediated shrinkage due to $K_{Ca}$ channel activation.
- Given Piezo1 widespread expression across animal cells, its contribution to volume regulation may represents a generalizable and conserved cell mechanism.

## Introduction

The regulation of cell volume is an evolutionarily conserved process essential for maintaining cell viability and function. Beyond counteracting environmental osmotic perturbations, it also underpins key physiological processes, such as proliferation, migration and cell death (Hoffmann et al., 2009). Cell volume regulation is

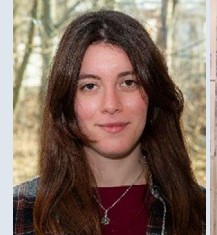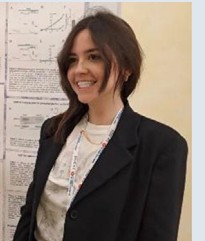

**Maria Vittoria Leonardi** is a PhD student at the Laboratory of Molecular Physiology, University of Perugia. Her research focuses on investigating the biophysical properties of ion channels using electrophysiology and computational approaches, including molecular dynamics simulations, with a particular emphasis on their role in cancer aggressiveness. **Angela Di Battista** is a research fellow at the Laboratory of Molecular Physiology, University of Perugia. Her research focuses on the biophysical characterization of ion channels through electrophysiology, with particular attention to their role in shaping the tumour microenvironment and promoting cancer aggressiveness.

fundamentally achieved by controlling water flux across the plasma membrane, driven by osmotic gradients through the activity of ion channels and transporters thatmove the principal osmolytes, such as$Na^+$, $K^+$ and $Cl^-$). Experimentally, the ability of cells to regulate their volume is often assessed by measuring the regulatory volume decrease (RVD), a complex homeostatic process whereby cells gradually restore their original volume following the rapid swelling induced by exposure to a hypotonic solution. RVD involves the coordinated activation of $Cl^-$ and $K^+$ channels, resulting in the net efflux of KCl and consequent osmotic loss of water. Extensive literature identifies the volume-regulated anion channel (VRAC), which mediates the swelling-activated $Cl^-$ current ($I_{Cl,swell}$), as the primary pathway for $Cl^-$ efflux during RVD in most animal cells (Formaggio et al., 2019; Hoffmann et al., 2015; Sardini et al., 2003; Sforna et al., 2017, 2022). In contrast, the pathways responsible for $K^+$ efflux are less well defined and appear to be highly cell type-dependent. For instance, a variety of $K^+$ channels have been implicated, including stretch-activated (Dubinsky et al., 2000; Filipovic & Sackin, 1992; Sackin, 1989), voltage-dependent (Cahalan & Lewis, 1988; Felipe et al., 1993), and $Ca^{2+}$-activated $K^+$ ($K_{Ca}$) channels of large (BK), intermediate (IK), and small (SK) conductance (Park et al., 1994; Wang et al., 2003; Weskamp et al., 2000), all of which have been shown to contribute to volume regulation in various cell types.

In addition to the osmotic water flux induced by transmembrane ion movement, cell volume regulation also depends on the cells ability to sense mechanical forces at the plasma membrane. Piezo channels, including Piezo1 and Piezo2, are the primary mechanosensitive channels in vertebrates and are permeant to both monovalent and divalent cations (Coste et al., 2010). Piezo2 is predominantly expressed in sensory neurons, whereas Piezo1 is widely expressed across nearly all cell types (Ranade et al., 2014; Retailleau et al., 2015). Piezo1 is activated by various mechanical stimuli, including membrane stretch resulting from cell swelling, and mediates $Ca^{2+}$ influx in response to hypotonic stress in multiple cell types. Studies conducted in recent years have begun to elucidate the emerging role played by Piezo1 in cell volume regulation (Cahalan et al., 2015; Johnson et al., 2024; Michelucci & Catacuzzeno, 2024; Michelucci et al., 2023; Sforna et al., 2022).

Significant insights into the mechanisms of cell volume regulation have emerged from studies on tumour cells, particularly glioblastoma (GBM), the most common and aggressive form of brain tumour, marked by a poor prognosis and a median survival of approximately 15 months (Davis, 2018; Davis et al., 1998). GBM cells represent a valuable and useful tool due to their remarkable ability to modulate cell volume in support to key processes such as migration, invasion and resistance to cell death. Several ion channels have been implicated in mediating these volume-dependent processes (Catacuzzeno et al., 2021; Edalat et al., 2016; Guerriero et al., 2024; Klumpp et al., 2018; Serpe et al., 2022; Sforna et al., 2017; Watkins & Sontheimer, 2011). We recently identified a specific mechanism in human U87-MG cells whereby $Ca^{2+}$ entry through mechanosensitive channels, triggered by hypotonic-induced cell swelling, plays a critical role in activating both IK and BK channels, ultimately enabling RVD response. Notably, activation of both $K_{Ca}$ channels is also observed following Yoda1 application under isotonic conditions, suggesting that the hypotonic-induced $Ca^{2+}$ influx is mediated by Piezo1 activation (Michelucci et al., 2023). However, the lack of highly selective pharmacological inhibitors for Piezo1 precludes definitive exclusion of contributions from other mechanosensitive $Ca^{2+}$-permeable channels. Although the well-characterized peptide GsMTx4 effectively inhibits Piezo1 activity, its limited specificity remains a concern, as it also modulates other mechanosensitive cation channels, including members of the TRP family (Alessandri-Haber et al., 2009; Bowman et al., 2007; Spassova et al., 2006). Thus, while GsMTx4 is a valuable tool for probing mechanosensitive channel function, potential off-target effects necessitate cautious interpretation of experimentaò outcomes. In this context, genetic approaches are essential for definitively linking Piezo1 activity to specific cell responses, and for excluding contributions from alternative mechanotransduction pathways.

To definitively assess the role of Piezo1 in cell volume regulation in GBM cells, we generated a stable Piezo1 knock-out (Piezo1 KO) U87-MG cell line using the CRISPR-Cas9 gene editing technology. Our findings demonstrate that Piezo1 is the primary mechanosensitive channel required for hypotonic-induced RVD in U87-MG cells. This process involves $Ca^{2+}$-dependent activation of the two predominant $K_{Ca}$ channels in these cells (IK and BK) that, together with Piezo1-independent activation of VRAC, mediates the KCl efflux essential for cell volume recovery. Notably, Piezo1 activation by Yoda1 under isotonic conditions induces a biphasic response characterized by an initial cell swelling driven by $Na^+$ entry, followed by a shrinkage driven by the $Ca^{2+}$-dependent activation of $K_{Ca}$ channels. This response underscores Piezo1 ability to modulate cell volume independently of osmotic perturbations or other $Ca^{2+}$-dependent pathways. Given the widespread expression of Piezo1 across animal cells, these findings suggest a diffuse role for Piezo1 in cell volume regulation and mechanotransduction.

## Methods

### Cell culture

Engineered Piezo1 KO and wild-type (WT) U87-MG cell lines (ICLC catalog code: HTL00013) were maintained in tissue culture polystyrene flasks using High Glucose Dulbecco's Modified Eagle Medium (DMEM; Euroclone S.p.A., Pero, Milan, Italy) supplemented with 10% heat-inactivated fetal bovine serum (FBS, Euroclone), 100 IU/ml penicillin G (1%), 100 µg/ml streptomycin (1%), and 1 mM sodium pyruvate. Cultures were incubated at 37°C in a 5% $CO_2$-humidified atmosphere. The medium was changed twice a week, and the cells were sub-cultured upon reaching confluence. For experiments, cells were seeded in Petri dishes at a density of 30,000–50,000 cells/ml. Electrophysiological recordings were performed at 2 or 3 days after seeding.

### Generation of the U87-MG Piezo1 KO cell line through the CRISPR/Cas9 approach

The U87-MG Piezo1 KO cell line was generated using a CRISPR/Cas9 double-cut strategy. Briefly, the PX459 V2.0 vector (pSpCas9(BB)-2A-Puro; Addgene Plasmid no. 62988), which co-expresses Cas9, a puromycin resistance gene, and a single-guide RNAs (sgRNA), was used to construct two plasmids targeting the third exon of the *PIEZO1* gene (sgRNA1 5′-GUCACACAGGCCGCCU CCUG-3′ and sgRNA2 5′-GCCCCGCCUGGACCA GCUCC-3′). The dual-vector approach was designed to induce a 31 bp deletion Cas9-mediated double-stranded DNA cleavage. U87-MG cells at 90% of confluence in a 100 $mm^2$ dish were transfected with a mixture of 15 µg of each plasmid using Lipofectamine 3000 (Thermo Fisher) according to the manufacturer's instructions. Forty-eight hours post-transfection, cells were selected with 1 µg/ml of puromycin for 72 hours and then plated in 96-well plates for clonal isolation. Genomic DNA from individual clones was screened by PCR using primers (MAN1449; MAN1451) flanking the target region to identify the clone carrying the expected 31 bp deletion in the third exon of the *PIEZO1* gene (Fig. 1). This deletion introduced a frameshift after amino acid 56 in the coding region, resulting in a truncated protein comprising the first 56 WT residues followed by 38 aberrant amino acids before a premature stop codon (Fig. 1). As this truncated protein comprises only a small fraction of the full-length ~2500 amino acids Piezo1 protein, it is highly unlikely to be stable. Such aberrant proteins are typically targeted for rapid degradation by nonsense-mediated mRNA

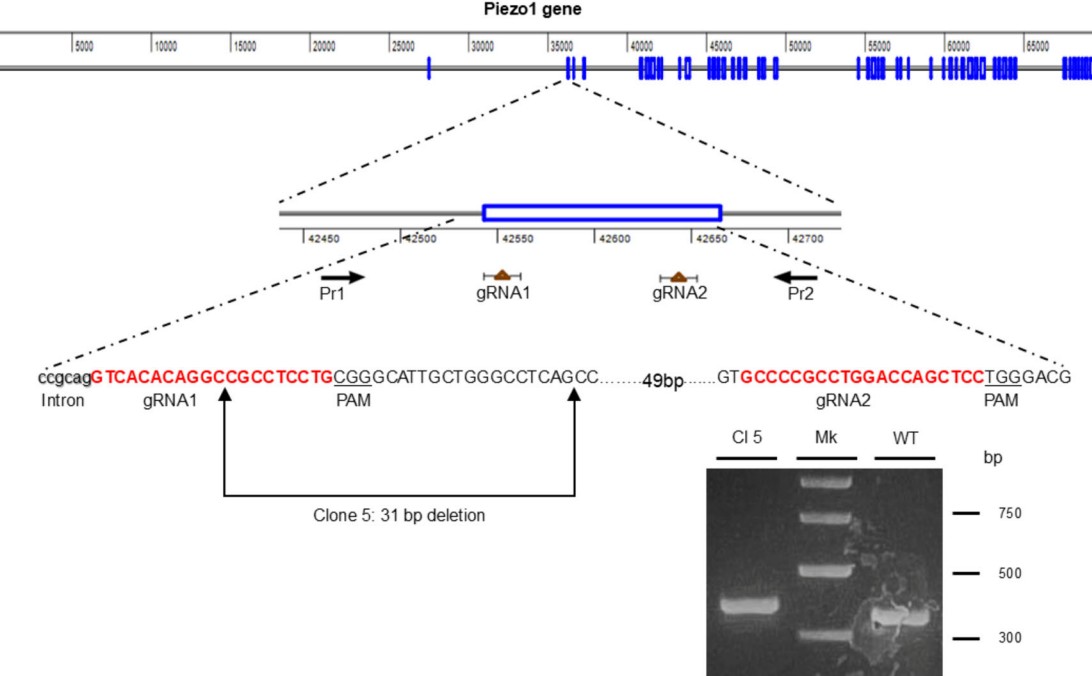

**Figure 1. Experimental approach for the generation of the U87-MG Piezo1 KO cell line**
Top, schematic representation of the CRISPR/Cas9-based strategy used to abolish *PIEZO1* gene expression. Bottom, PCR analysis of genomic DNA from Clone 5 (ΔP5) to assess *PIEZO1* deletion. Primers 1 (5′-TGGGGGCTATTGTCCGGCTGAGCG-3′) and 2 (5′ TCCCTGACCCCCAGGATGGCTATC-3′) were used to amplify a 257 bp region of the *PIEZO1* gene encompassing the two Cas9 target sites. The PCR product from Clone 5 was 31 bp shorter than that of the WT control, consistent with successful deletion. Sanger sequencing of the PCR amplicons confirmed the expected genomic excision.

decay control pathways, thereby preventing expression of functional proteins. Therefore, significant levels of the truncated protein are not expected to be present.

## SDS-PAGE electrophoresis and western blot analysis

WT and Piezo1 KO U87-MG cells were resuspended in 10 mM sodium phosphate buffer (pH 6.0) containing 0.1% (v/v) Nonidet NP-40 (Sigma-Aldrich) and 0.3% (v/v) taurodeoxycholate (Sigma-Aldrich). Cell lysis was achieved by three rounds of sonication (30 s each) on ice. All procedures were performed at 4°C. Protein concentrations were determined using the Bradford assay. Equal amounts of protein (30 µg per sample) were separated by SDS-PAGE (7.5% acrylamide-bisacrylamide) and transferred to nitrocellulose membranes. Membranes were incubated overnight at 4°C with a polyclonal anti-Piezo1 antibody (Thermo Fisher Scientific, PA5-116998; dilution 1:500), followed by a 1-h incubation at room temperature with an HRP-conjugated anti-rabbit IgG secondary antibody (Cell Signalling Technology). Immunoreactive bands were visualized using the Clarity Western ECL substrate (Bio-Rad Laboratories). Densitometric analysis was performed using iBrigth

Analysis software CL1500 (Invitrogen, Thermo Fisher Scientific, Waltham, MA, USA), with band intensities normalized to $\beta$-actin as an internal reference. Data were obtained from three independent cell cultures. A representative, unedited uncropped gel image with two independent cell preparations per each group is shown in Fig. 2.

## Patch-clamp electrophysiology

The whole cell dialysed configuration was used for electrophysiological recordings. Currents and voltages were amplified with a HEKA EPC-10 amplifier (List Medical), and analysed with the software Patch-Master package (version 2_60, ELEKTRONIK). For on-line data collection, macroscopic currents were filtered at 3 kHz and sampled at 50 µs/point. The extracellular Ringer solution contained: NaCl 140 mM, KCl 5 mM, $CaCl_2$ 2 mM, $MgCl_2$ 2 mM, MOPS 5 mM, glucose 10 mM (pH 7.40). The intra-cellular solution contained: KCl 155 mM, EGTA-K 1 mM, MOPS 5 mM, $MgCl_2$ 1 mM (pH 7.20). The desired free $Ca^{2+}$ concentration (60 nMM) was obtained by adding an amount of $CaCl_2$ (calculated with the webmax software: www.stanford.edu/~cpatton/webmaxc/webmaxcS.htm)

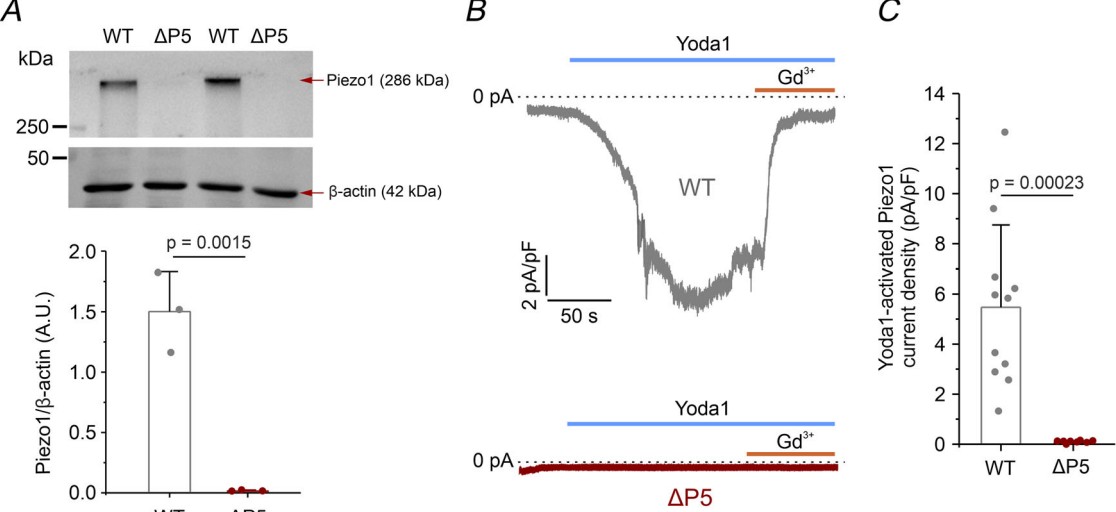

**Figure 2. Validation of U87-MG Piezo1 KO cell model**
*A*, top: representative immunoblots showing Piezo1 protein expression (upper gel) in lysates from WT (*n* = 3 independent cell cultures) and ΔP5 (*n* = 3 independent cell cultures) cells. Predicted molecular weights of Piezo1 (286 kDa, upper gel) and $\beta$-actin (42 kDa, loading control; lower gel) are indicated by red arrows. Bottom: quantification of Piezo1 band intensity normalized to $\beta$-actin. Data are shown as mean ± SD; statistical significance was assessed with two-sample unpaired *t* test. *B*, representative whole-cell current traces recorded in continuous voltage-clamp mode at the holding potential of −80 mV, in WT (top, dark grey trace) and ΔP5 (bottom, red trace) cells. Currents were evoked by application of isotonic solution containing 10 µM Yoda1 (blue bar) followed by 10 µM $Gd^{3+}$ (orange bar). All recordings were performed in the constant presence of 10 µM DCPIB, 3 µM TRAM-34 and 1 µM paxilline, blockers of VRAC, IK and BK channels, respectively. *C*, bar plot showing the average current density (pA/pF), calculated by subtracting the current in the presence of $Gd^{3+}$ from the peak Yoda1-evoked current, in WT (grey bar, *n* = 11 cells from three independent cultures) and ΔP5 (red bar, *n* = 8 cells from three independent cultures) cells. Data are shown as mean ± SD; statistical significance was assessed using an unpaired two-sample *t* test. Individual data points for each experimental group are reported.

to the internal solution. Patch electrodes with resistances 3–5 MΩ were fabricated on a PUL-100 Micropipette Puller (World Precision Instruments) from borosilicate glass (OD1.5 mm, ID 0.84 mm, World Precision Instruments). Access resistances ranged between 8 and 15 MΩ and were actively compensated to ~50%. The hypotonic solution was prepared by adding 30% distilled water to the extracellular Ringer solution. All reagents were fresh daily solubilized at the concentrations stated, and bath applied with a gravity perfusion system. Octanol (1 mM) was added to extracellular Ringer solution to block the gap-junctions (Eskandari et al., 2002). All chemicals used were of analytical grade. Dimethyl sulfoxide (DMSO) and gadolinium chloride ($GdCl_3$) were purchased from Sigma. 4-[(2-butyl-6,7-dichloro-2-cyclopentyl-2,3-dihydro-1-oxo-1H-inden-5-yl)oxy] butanoic acid (DCPIB), TRAM-34, paxilline, and Yoda1 were purchased from Tocris Biosciences. $GdCl_3$ was dissolved in water at the stock concentration of 100 mM and used at the final concentration of 10 μM. DCPIB, TRAM-34, paxilline, and Yoda1 were dissolved in DMSO at stock concentrations of 10, 6, 10 and 10 mM, and used at final concentrations of 10, 3, 1 and 10 μM, respectively. The maximal DMSO concentration in the recording solutions was 0.1%. All recordings were carried out at room temperature (18–22°C).

## Videoimaging for cell volume assessment

Experiments were carried out using the same extracellular Ringer solution used for electrophysiological experiments, but without octanol. To assess hypotonic-induced cell swelling, cells were perfused with the 30% hypotonic solution prepared using the same procedure described in 'Patch-clamp electrophysiology' section. Cell volume changes were monitored by measuring the relative projection area of cells using a time-laps videoimaging technique. Cells were visualised through a ×40 objective lens on an inverted microscope connected to a video camera (Axiocam/Cm1, Zeiss). Images were acquired every 2 min over a total recording period of 60 min, and saved as TIFF files. Projected cell area was quantified using ImageJ. Relative cell area ($A_{rel}$) was calculated at each time point ($A_t$) upon exposure to hypotonic solution by normalising to the baseline area ($A_0$), defined as the average of the three time points recorded in the first 4 min under isotonic conditions. The extent of RVD was measured 60 min after the onset of the regulatory response, which began shortly after cell reached the peak of swelling under hypotonic solution, and was calculated as follows:

$$\%RVD = (A_{rel,peak} - A_{rel,60})/(A_{rel,peak}) \times 100,$$

where $A_{rel,peak}$ and $A_{rel,60}$ are the relative projected cell areas at the peak of swelling and at the end of the experiment, respectively (Michelucci et al., 2023; Sforna et al., 2017, 2022). In each experimental session, typically fewer than five cells were monitored and only those maintaining a round shape throughout the experiment were included in the analysis. WT cells used as controls for the Piezo1 KO group were the same as those used for comparisons across all experimental conditions (e.g. 0Ca, 0Na, TRAM-34, paxilline), and were recorded within the same experimental sessions. For each condition tested, at least one WT control was acquired in parallel on the same day, ensuring that all comparisons were conducted under identical conditions. Each experimental condition included a minimum of three biological replicates from independent cultures.

## Statistical analysis

Statistical analyses were conducted using Origin 8.0 software. Data are expressed as mean ± SD. Normality was assessed using the Shapiro–Wilk test. Comparisons between two independent groups with normal distribution were made using unpaired two-sample *t* test. For comparison involving multiple groups, one-way analysis of variance (ANOVA) followed by Tukey's *post hoc* test was applied. Non-normally distributed data were analysed using the Mann–Whitney *U* test was used. Exact *P* values are reported, and statistical significance was set at $P < 0.05$. Sample sizes ($n$) are provided in the figure legends. All experiments were independently repeated at least three times to ensure reproducibility.

## Results

### Generation and validation of an engineered U87-MG Piezo1 KO cell model

U87-MG cells with stable KO for the *PIEZO1* gene were generated using the CRISPR/Cas9 gene editing (details provided in the Methods section and Fig. 1). Among several clones, we selected Clone 5 (ΔP5), which harbours a biallelic deletion of 31 bp in the third exon of the *PIEZO1* gene. This deletion caused a frameshift after amino acid 56 in the coding region, resulting in a truncated protein comprising the first 56 WT residues followed by 38 aberrant amino acids before a premature stop codon (Fig. 1). The truncated product retains only the initial transmembrane domain and is therefore incapable of assembling into a functional channel. Given its minimal length, it is also unlikely to be stable. To confirm the genetic ablation of Piezo1, we performed western blot analysis using a commercial anti-Piezo1 polyclonal antibody which targets a C-terminal epitope (amino acids

2230–2420 of human *PIEZO1*; see Methods for details). Robust Piezo1 expression was detected in WT cells as a strong band near the expected molecular weight ($\sim$286 kDa), whereas no protein signal was observed in $\Delta$P5 cells, confirming successful gene disruption (Fig. 2*A*). Given the frameshift mutation and the absence of detectable protein using our polyclonal antibody, we are confident that no Piezo1 protein, including the N-terminal 56 amino acid domain, is present in the KO cell line.

To further validate the functional loss of Piezo1 protein, whole-cell patch-clamp recordings were performed to assess Piezo1-mediated currents. Since Piezo1 is a mechanosensitive channel permeable to cations, namely $Na^+$ and $Ca^{2+}$, inward currents were recorded using a continuous voltage-clamp protocol with cells held at $-80$ mV to establish a strong electrochemical gradient for both cations. Piezo1 activation was induced by perfusing an isotonic solution containing 10 $\mu$M Yoda1, a potent and selective Piezo1 agonist (Syeda et al., 2015). To confirm specificity, 10 $\mu$M $Gd^{3+}$, a non-specific mechanosensitive channel blocker, was applied to inhibit the Yoda1-induced current. Recordings were conducted in the continuous presence of 10 $\mu$M DCPIB, 3 $\mu$M TRAM-34 and 1 $\mu$M paxilline, to block VRAC, IK and BK channels, respectively, which are known to be activated downstream of Piezo1-mediated $Ca^{2+}$ influx in U87-MG cells (Michelucci et al., 2023).

In WT cells, acute application of Yoda1 elicited a significant inward $Gd^{3+}$-sensitive current with an average amplitude of 5.5 $\pm$ 3.3 pA/pF. Consistent with the absence of Piezo1 in $\Delta$P5 cells as confirmed by western blot, no Yoda1-induced current was detected in the KO clone (Fig. 2*B* and *C*).

These results validated the effectiveness of our CRISPR/Cas9 approach in generating a stable U87-MG Piezo1 KO cell line. Hereafter, the $\Delta$P5 clone is referred to as the Piezo1 KO group and compared against WT cells as controls.

## Piezo1 contributes to the RVD

We investigated the role of Piezo1 in the RVD, a well-established experimental model for studying cell volume regulation. Using phase-contrast microscopy, we performed time-laps videoimaging recordings and quantified changes in cell volume by measuring the relative projected cell area of spherical cells. Images were acquired every 2 min over a 60-min period. Following a 4-min control period under isotonic conditions (first three images), cells were exposed to a 30% hypotonic solution for the remaining 56 min (Michelucci et al., 2023; Sforna et al., 2022). The average time course of the relative cell area revealed marked differences between the two genotypes. Both WT and Piezo1 KO cells underwent rapid swelling upon exposure to the hypotonic solution; however, Piezo1 KO cells exhibited a greater swelling amplitude, consistent with impaired activation of the RVD process. This observation aligns with our previous findings in engineered HEK293T cells, where an inverse relationship between the extent of cell swelling and Piezo1 expression was observed (Sforna et al., 2022). Furthermore, while WT cells fully restored their original volume following the hypotonic challenge, Piezo1 KO cells displayed a significantly impaired RVD response (Fig. 3*A* and *B*). Specifically, the extent of RVD was 98.9 $\pm$ 36.0% for WT cells, compared to only 34.9 $\pm$ 26.1% for Piezo1 KO cells (Fig. 3*C*). Notably, the extent of volume recovery in Piezo1 KO cells was comparable to that observed in WT cells following selective pharmacological inhibition of either IK or BK channels, or when extracellular $Ca^{2+}$ entry through mechanosensitive channels was prevented (Michelucci et al., 2023). Collectively, these data highlight the essential role of Piezo1 in driving RVD in U87-MG cells.

## Piezo1 is essential for hypotonic-induced $K_{Ca}$ channels activation

We recently demonstrated that both hypotonic stimulus and selective Piezo1 agonist Yoda1 activate $K_{Ca}$ channels via extracellular $Ca^{2+}$ influx (Michelucci et al., 2023). However, the direct involvement of Piezo1 in the hypotonic-induced activation remained inconclusive, due to the lack of highly selective Piezo1 inhibitors (Alessandri-Haber et al., 2009; Bowman et al., 2007; Spassova et al., 2006). To overcome this limitation, we employed our stable U87-MG Piezo1 KO cell line to assess IK and BK channel activation in response to hypotonic conditions.

To independently assess IK and BK currents, we employed selective pharmacological blockers in our recording solutions. The IK current was isolated by supplementing the extracellular solution with 10 $\mu$M DCPIB and 1 $\mu$M paxilline. Consistent with our previous findings (Michelucci et al., 2023), exposure of WT cells to a 30% extracellular hypotonic solution induced a robust outward current, measured at 0 mV from 1-s current ramps. This current exhibited the characteristic biophysical and pharmacological properties of IK channels, including voltage independence, a reversal potential near the $K^+$ equilibrium potential ($\sim -80$ mV), and sensitivity to TRAM-34 (Fig. 4*A*, top). In contrast, the same hypotonic stimulus failed to effectively activate a TRAM-34-sensitive current in Piezo1 KO cells, (Fig. 4*A*, bottom). Quantitative analysis revealed that the average hypotonic-induced IK current density was more than 10-fold greater in WT cells (4.7 $\pm$ 2.3 pA/pF) compared to Piezo1 KO cells (0.3 $\pm$ 0.2 pA/pF) (Fig. 4*B*).

To exclude the possibility that the absence of IK current activation in response to hypotonic stimulation in Piezo1 KO cells was due to reduced IK channel expression, we measured maximal IK current under conditions designed to ensure full activation. Cells were dialysed with a pipette solution containing a high free $Ca^{2+}$ concentration (400 nM) and exposed to the IK channel opener NS309 (Fig. 4C). Under these conditions, maximal IK current density in Piezo1 KO cells (7.7 ± 4.2 pA/pF) was nearly identical to that in WT cells (7.0 ± 3.6 pA/pF) (Fig. 4D). These findings confirm that IK channel expression and functional capacity are preserved in Piezo1 KO cells, indicating that the loss of hypotonic-induced IK activation is entirely attributable to the absence of Piezo1-mediated $Ca^{2+}$ influx.

We next assessed BK channel activity, by isolating BK currents using an extracellular application of 10 μM DCPIB and 3 μM TRAM-34. To quantify BK current activation by hypotonic stress, we applied 100 ms depolarizing voltage steps from −60 to 140 mV (20 mV increments) from a holding potential of −40 mV. Under isotonic conditions, this protocol elicited an outward paxilline-sensitive current in both WT and Piezo1 KO cells, characterized by high noise at voltages greater than 50 mV, typical of currents passing through the high-conductance BK channel (Fig. 5A and B). Quantitative analysis revealed no significant difference in BK current density between the two genotypes under

isotonic conditions, with comparable values at 100 mV (30.4 ± 23.4 pA/pF and 33.5 ± 10.9 pA/pF for the WT and Piezo1 KO, respectively) and 120 mV (49.1 ± 23.1 pA/pF and 54.9 ± 15.5 pA/pF for the WT and Piezo1 KO, respectively), indicating that Piezo1 deficiency did not affect BK channel expression (Fig. 5C). In contrast, a significant difference emerged under hypotonic conditions. In WT cells, the BK current was robustly activated even at negative, physiologically relevant voltages, with average current densities of 5.2 ± 6.0 pA/pF at −40 mV and 16.3 ± 15.5 pA/pF at −20 mV. No detectable hypotonic-activated current was observed at these voltages in Piezo1 KO cells (Fig. 5D). Notably, a small but significant hypotonic-induced current activation was detected at higher (>50 mV) depolarizations, suggesting a minimal Piezo1-independent component of BK channel activation (Fig. 5B).

Our results demonstrate that activation of both $K_{Ca}$ channels in response to hypotonic stress depends on $Ca^{2+}$ influx through Piezo1. To further confirm direct regulation of $K_{Ca}$ channels by Piezo1, we performed whole-cell patch-clamp recordings under isotonic conditions using the selective Piezo1 agonist Yoda1 (10 μM). Consistent with our previous findings in this cell line (Michelucci et al., 2023), Yoda1 robustly activated both IK (Fig. 6A and B) and BK (Fig. 6C and D) currents in WT cells, closely replicating the responses observed under hypotonic stimulation. In contrast, no activation

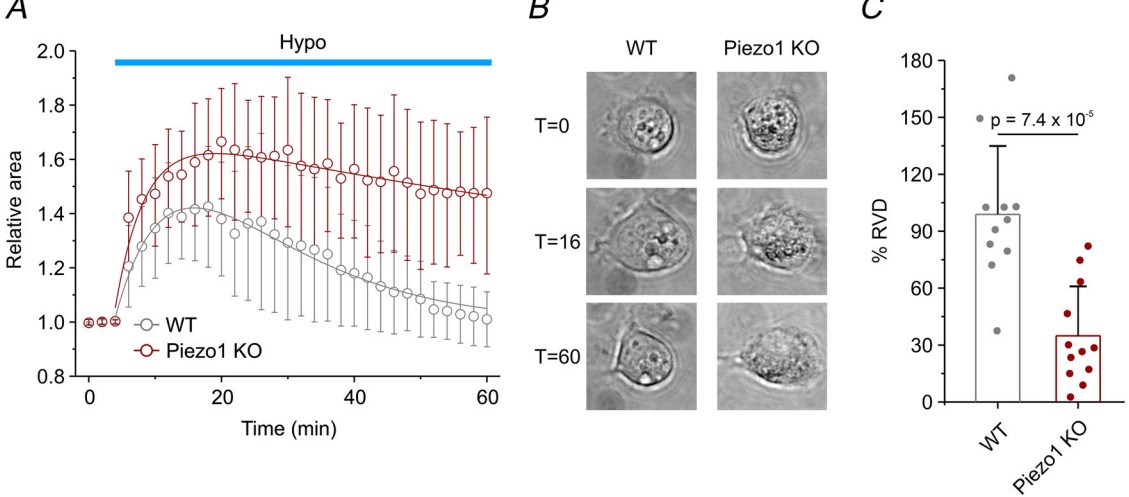

**Figure 3. Assessment of RVD during hypotonic stress in WT and Piezo1 KO U87-MG cells using video-imaging analysis**

*A*, time course of RVD, measured as changes of the relative projected cell area under isotonic conditions (first three time points, 4 min) and during exposure to a 30% hypotonic solution (Hypo, cyan bar), in WT (grey circles, *n* = 11 cells from four independent cultures) and Piezo1 KO (red circles, *n* = 12 cells from four independent cultures) cells. Data are shown as mean ± SD. *B*, representative phase-contrast images acquired at baseline (*t* = 0 min) and during exposure to 30% hypotonic solution at varying time points (*t* = 16 and *t* = 60 min), in WT (left images) and Piezo1 KO (right images) cells. *C*, bar plot showing the percentage of RVD (%RVD) in the two different genotypes. Data are shown as mean ± SD; statistical significance was assessed using an unpaired two-sample *t* test. Individual data points for each experimental group are reported.

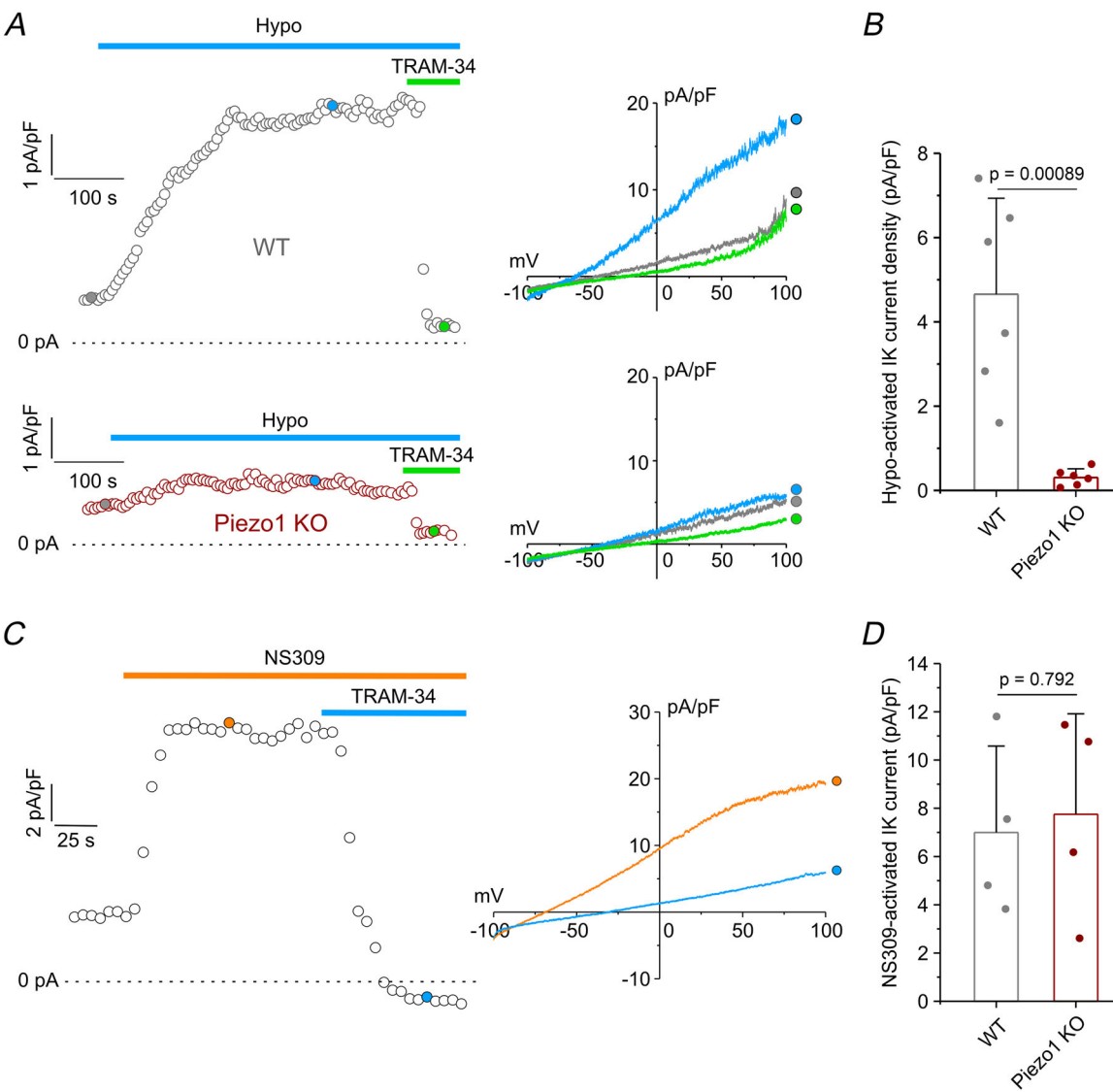

**Figure 4. Patch-clamp recordings of hypotonic- and NS309-activated IK currents in WT and Piezo1 KO U87-MG cells**

*A*, left: representative time courses of current density (pA/pF) measured at 0 mV from current ramps (shown on the right) in response to a 30% hypotonic solution (Hypo, cyan bar) and after the addition of the IK channel inhibitor TRAM-34 (3 μM, green bar). All recordings were performed in the presence of 10 μM DCPIB and 1 μM paxilline to block VRAC and BK channels, respectively. Right: representative current ramps obtained by applying 1-s voltage ramps from −100 to 100 mV, from a holding potential of −40 mV, under control isotonic conditions (grey traces), during application of hypotonic solution (cyan trace), and after the addition of TRAM-34 (green trace). Top and bottom panels correspond to WT and Piezo1 KO U87-MG cells, respectively. *B*, bar plot showing mean current density (pA/pF) at 0 mV, calculated as the difference between the current recorded during hypotonic stimulation and the residual current in the presence of TRAM-34, in WT ($n = 6$ cells from three independent cultures; grey bar) and Piezo1 KO ($n = 6$ cells from three independent cultures; red bar) cells. Data are shown as mean ± SD; statistical significance was assessed with a two-sample unpaired $t$ test. *C*, left: representative time course of IK current density (pA/pF) measured from current ramps at 0 mV (shown on the right) in a WT cell, showing activation by 10 μM NS309 (orange bar) and subsequent inhibition by 3 μM TRAM-34 (cyan bar). Right: representative 1-s current ramps from −100 to 100 mV from a holding potential of −40 mV, during application of NS309 (orange trace) and TRAM-34 (cyan trace). *D*, bar plot showing the average IK current density at 0 mV, calculated as the difference between the current activated by NS309 and the residual current following TRAM-34 addition, in WT ($n = 4$ cells from three independent cultures) and Piezo1 KO ($n = 4$ from three independent cultures) cells. Data are shown as mean ± SD; statistical significance was assessed with an unpaired two-sample $t$ test. Individual data points for each experimental group are reported.

of either current was detected in Piezo1 KO cells, further confirming that the $Ca^{2+}$-dependent activation of these $K_{Ca}$ channels is specifically mediated by Piezo1.

## Piezo1 does not affect the functional expression of the VRAC-mediated $I_{Cl,swell}$

We previously demonstrated that in HEK293T cells, Piezo1 regulates cell volume by modulating the $I_{Cl,swell}$ mediated by VRAC through the $Ca^{2+}$-induced $Ca^{2+}$ release (CICR) mechanism (Sforna et al., 2022). This $Ca^{2+}$-dependent enhancement of $I_{Cl,swell}$ is essential for full VRAC activation and effective execution of RVD. In GBM cells, $I_{Cl,swell}$ via VRAC also plays a crucial role in

RVD (Sforna et al., 2017). However, unlike in HEK293T cells, in GBM cells its activation by hypotonic stress occurs independently of intracellular $Ca^{2+}$ (Catacuzzeno et al., 2014). To confirm this observation, we assessed hypotonic-induced $I_{Cl,swell}$ in the absence of Piezo1. VRAC-mediated $I_{Cl,swell}$ was recorded using 1-s voltage ramps from $-100$ to $100$ mV (from a holding of $-40$ mV) and assessed near the $K^{+}$ equilibrium potential ($-80$ mV) to minimize contributions of $K_{Ca}$ currents. Additionally, any potential interference from $K_{Ca}$ channels was prevented by the continuous presence of 3 μM TRAM-34 and 1 μM paxilline in the extracellular solution. As shown by representative time courses, application of the hypotonic solution elicited a marked increase in inward DCPIB-sensitive current in both WT and Piezo1 KO

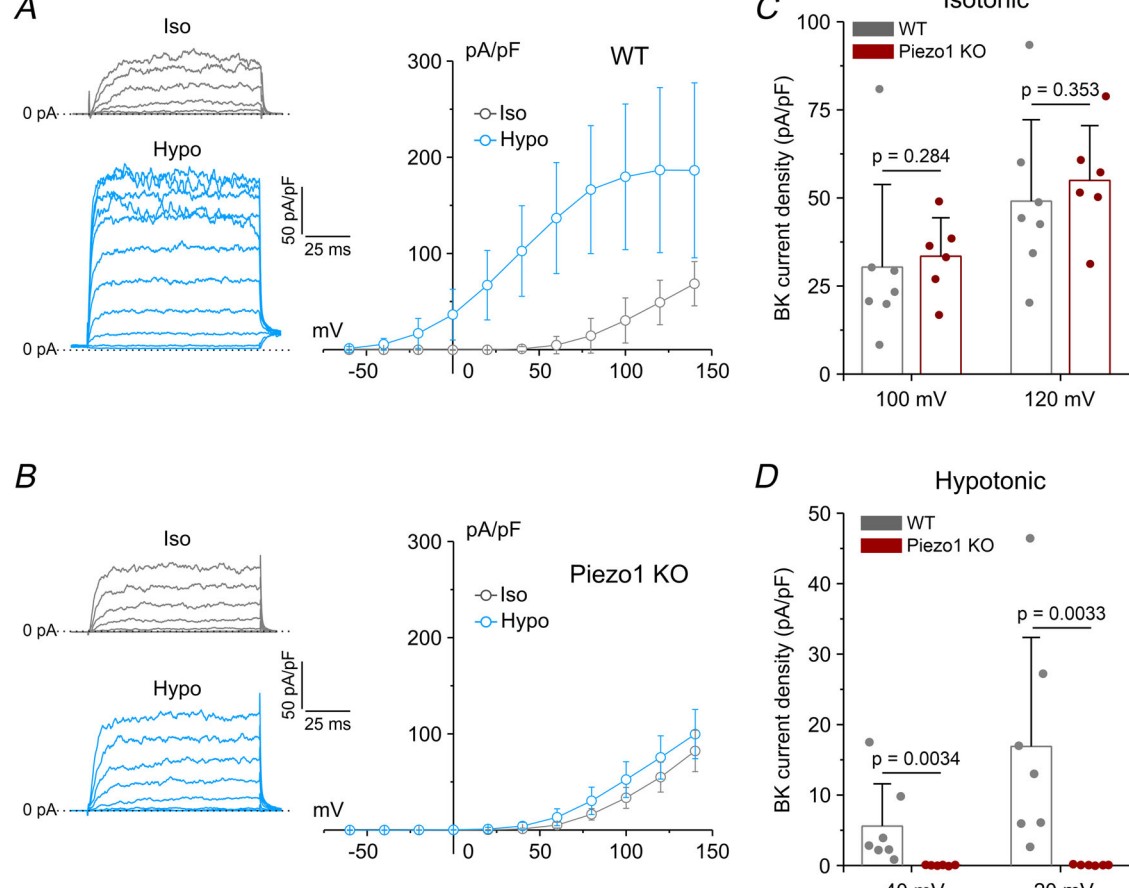

**Figure 5. Patch-clamp recordings of hypotonic-activated BK current in WT and Piezo1 KO U87-MG cells**
*A* and *B*, left: representative families of current traces recorded from WT (*A*) and Piezo1 KO (*B*) cells using 100 ms voltage steps from $-60$ to 140 mV (20 mV increments), from a holding potential of $-40$ mV, under isotonic conditions (Iso; grey traces) and during application of a 30% hypotonic solution (Hypo; blue traces). Right: corresponding average current–voltage (*I–V*) relationships under control isotonic (grey circles) and hypotonic (blue circles) conditions, in WT (*A*, *n* = 7 cells from three independent cultures) and Piezo1 KO (*B*, *n* = 6 cells from three independent cultures) cells. Data are shown as mean ± SD. *C* and *D*, bar plots showing the average BK current density (pA/pF) at two high depolarized voltages (100 and 120 mV) under isotonic control conditions (*C*) and at two physiologically relevant hyperpolarized voltages ($-40$ and $-20$ mV), during hypotonic stimulation (*D*) in WT (grey bars) and Piezo1 KO (red bars) cells. Data are shown as mean ± SD; statistical significance was assessed using the non-parametric Mann–Whitney *U* test. Individual data points for each experimental group are reported.

cells. This current displayed the characteristic biophysical features of VRAC-mediated $I_{Cl,swell}$, including a reversal potential near the of $Cl^-$ equilibrium potential, moderate outward rectification, and time- and voltage-dependent inactivation (Fig. 7*A* and *B*). Quantitative analysis showed that the current density at −80 mV was essentially identical in WT (59.1 ± 25.8 pA/pF) and Piezo1 KO (61.2 ± 21.4 pA/pF) cells (Fig. 7*C*), indicating that VRAC expression and function remain unaffected by Piezo1 deletion.

To further confirm that VRAC activity in U87-MG cells is independent of $Ca^{2+}$ influx through Piezo1 following mechanical membrane stress, we applied 10 μM Yoda1 under isotonic conditions while recording $I_{Cl,swell}$ in the presence of TRAM-34 and paxilline, at both depolarized (50 mV) and hyperpolarized (−80 mV) potentials. Representative traces from WT cells (Fig. 7*D*) showed no detectable current in response to Yoda1 at either potential. In contrast, hypotonic challenge evoked a robust DCPIB-sensitive current exhibiting characteristic biophysical features of VRAC-mediated $I_{Cl,swell}$. This lack of VRAC activation by Yoda1 was further confirmed by quantitative analysis of current density at −80 mV (Fig. 7*E*). Collectively, these data indicate that VRAC activation occurs independently of Piezo1 activity.

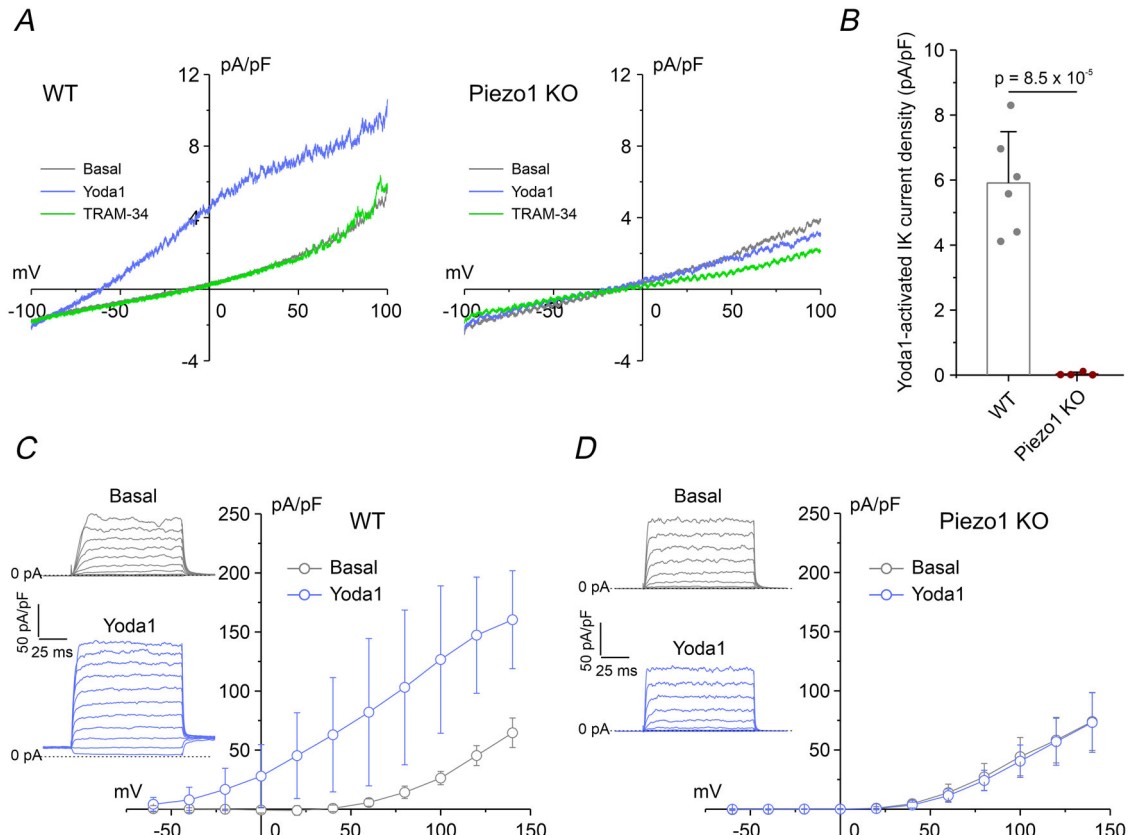

**Figure 6. Patch-clamp recordings of Yoda1-activated $K_{Ca}$ current**
*A*, representative current ramps recorded in WT (left) and Piezo1 KO (right) cells and obtained by applying 1-s voltage ramps from −100 to 100 mV, from a holding potential of −40 mV, during application of an isotonic solution containing 10 μM Yoda1 (purple trace) and following the addition of the IK channel inhibitor TRAM-34 (3 μM, green bar). The extracellular solution contained 10 μM DCPIB and 1 μM paxilline, blockers of VRAC and BK channel, respectively. *B*, bar plot showing mean IK current density (pA/pF) calculated as the difference between the current recorded in the presence of Yoda1 and the residual current after TRAM-34 application, in WT (*n* = 6 cells from three independent cultures, grey bar) and Piezo1 KO (*n* = 4 cells from three independent culture, red bar) cells. Data are shown as mean ± SD; statistical significance was assessed using an unpaired two-sample *t* test. Individual data points for each experimental group are reported. *C* and *D*, average current-voltage (*I–V*) relationships in WT (*C*, *n* = 7 cells from three independent cultures) and Piezo1 KO (*D*, *n* = 6 cells from three independent cultures) cells, under isotonic conditions in the absence (grey circles) and presence (purple circles) of 10 μM Yoda1. Data are shown as mean ± SD. Insets: representative families of current traces recorded using 100 ms voltage steps from −60 to 140 mV (20 mV increments), from a holding potential of −40 mV, under control conditions (grey traces) and following Yoda1 application (purple traces).

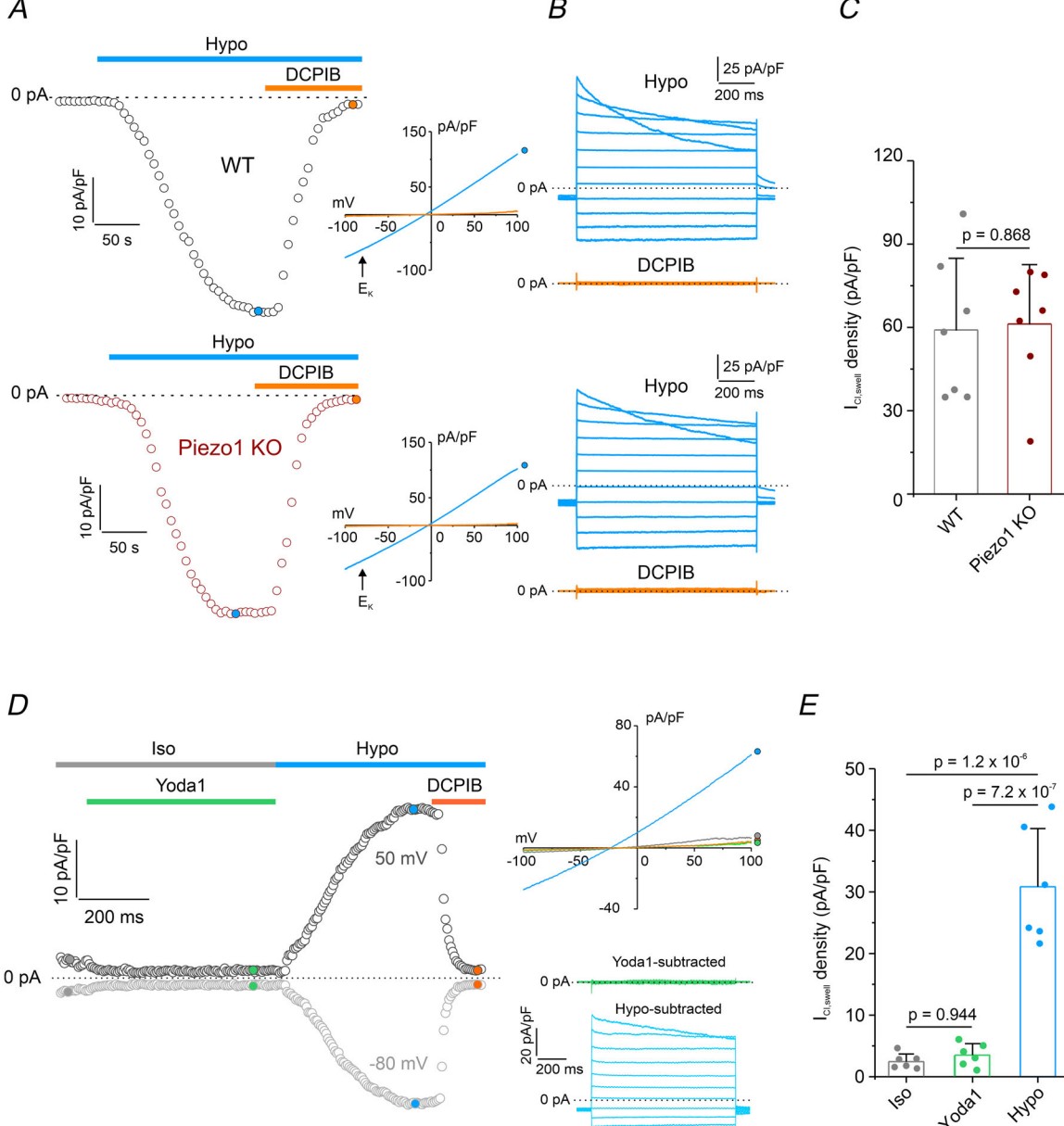

**Figure 7. Patch-clamp recording of VRAC-mediated $I_{Cl,swell}$ in WT and Piezo1 KO U87-MG cells**

*A*, time courses of current density (pA/pF) measured at −80 mV in WT (top) and Piezo1 KO (bottom) cells during application of a 30% hypotonic solution (Hypo; cyan bar) and following addition of 10 μM DCPIB (orange bar). Insets: representative 1-s current ramps from −100 to 100 mV, from a holding potential of −40 mV, recorded during exposure to hypotonic solution in the absence (cyan trace) or presence (orange trace) of 10 μM DCPIB. *B*, representative families of current traces evoked by applying 1-s voltage steps from −100 to 100 mV, in 20 mV increment (holding potential −40 mV) in the presence of a 30% hypotonic solution (cyan traces), and upon addition of 10 μM DCPIB (orange traces). *C*, bar plot showing average DCPIB-sensitive current density at −80 mV during exposure to hypotonic stimulation in WT (*n* = 7 cells from three independent cultures) and Piezo1 KO (*n* = 7 cells from three independent cultures) cells. Data are shown as mean ± SD; statistical significance was evaluated with an unpaired two-sample *t* test. *D*, left: time course of current density (pA/pF) at −80 mV (light grey circles) and +50 mV (dark grey circles) in a WT cell during sequential application of isotonic solution containing 10 μM Yoda1 (green bar), hypotonic solution alone (cyan bar), and hypotonic solution with 10 μM DCPIB (orange bar). Right, top: representative 1-s current ramps from −100 to 100 mV, from a holding potential of −40 mV, recorded under isotonic control (grey trace), isotonic + 10 μM Yoda1 (green trace), 30% hypotonic (cyan trace), and 30% hypotonic + DCPIB (orange) conditions. Right, bottom: representative families of current traces evoked by applying 1-s voltage steps from −100 to 100 mV (20 mV increments; holding potential −40 mV) under isotonic with 10 μM Yoda1 (green trace) or hypotonic (cyan trace) conditions, shown after subtraction of the DCPIB-insensitive

component. *E*, bar plot showing the average current density (pA/pF) at −80 mV under isotonic conditions (grey bar), in the presence of Yoda1 (green bar), and during hypotonic stimulation (cyan bar) in WT cells ($n$ = 6 cells from three independent cultures). Data are shown as mean ± SD; statistical significance was assessed using one-way ANOVA followed by *post hoc* Tukey's test. Individual data points for each experimental group are reported.

## Pharmacological activation of Piezo1 directly modulates cell volume

As shown in Fig. 3, Piezo1 is essential for hypotonic-induced RVD. However, the osmotic shock applied in our experiments (i.e. a 30% hypotonic solution) has limited physiological relevance, as such extreme conditions are unlikely to occur *in vivo*. Moreover, cells within tissues frequently regulate their volume to support processes such as proliferation or migration, even without osmotic imbalance. This raises the question of whether Piezo1 can modulate cell volume under isotonic conditions.

To investigate this aspect, we applied 10 μM Yoda1 to WT cells under isotonic conditions, following the experimental protocol used in Fig. 3. Remarkably, acute application of Yoda1 triggered a biphasic volume response: an initial swelling phase followed by a pronounced shrinkage, ultimately reducing cell volume below baseline levels. Consistent with Yoda1 specificity for Piezo1, no significant volume changes were observed in Piezo1-deficient cells (Fig. 8*A*), confirming that this response is directly mediated by Piezo1 activation. The biphasic volume response induced by Yoda1 likely reflects Piezo1 permeability to both $Na^+$ and $Ca^{2+}$ (Coste et al., 2010; Gnanasambandam et al., 2015). This supports the hypothesis that the initial swelling results from osmotic water influx driven by the sustained $Na^+$ entry into the cell, while the subsequent RVD is triggered by cytosolic $Ca^{2+}$ accumulation, leading to $K^+$ efflux through activated $K_{Ca}$ channels and the associated osmotic water loss.

To test this hypothesis, we first conducted experiments in the absence of extracellular $Na^+$, replacing it with the impermeant cation $NMDG^+$ to maintain electrical and osmotic balance. This approach also allowed us to assess the relative contributions of $Na^+$ and $Ca^{2+}$ to the observed volume changes. Under these conditions, Yoda1 application failed to induce cell swelling but instead triggered a marked cell shrinkage (Fig. 8*B*), confirming that the Yoda1-induced swelling is primarily driven by $Na^+$ influx through Piezo1, rather than by $Ca^{2+}$ entry, and is followed by osmotic water uptake.

Next, we tested the effect of removing extracellular $Ca^{2+}$. Under these conditions, Yoda1 induced a swelling comparable to the control conditions, providing a further evidence that $Ca^{2+}$ plays a minimal role in this initial phase. However, the recovery phase was markedly impaired, and no subsequent cell shrinkage occurred (Fig. 8*C*), likely due to the inability to activate $K_{Ca}$

channels. This interpretation is supported by experiments in which selective pharmacological inhibition of both IK and BK channels with TRAM-34 and paxilline, respectively, reproduced the same outcome observed in the absence of external $Ca^{2+}$ (Fig. 8*D*).

Together, these results demonstrate that Piezo1 regulates cell volume under isotonic conditions through a biphasic mechanism where $Na^+$ influx drives osmotic cell swelling, while subsequent $Ca^{2+}$-dependent activation of $K_{Ca}$ channels mediates cell shrinkage.

## Discussion

In this study, we investigated the role of Piezo1 in cell volume regulation in human U87-MG cells employing a stable CRISPR/Cas9-generated Piezo1 KO model. The effects of Piezo1 deletion on cell volume regulation are particularly evident in the process of RVD induced by hypo-osmotic shock. In particular, Piezo1-deficient cells exhibited a markedly impaired ability to recover their original volume after hypotonic-induced initial swelling, in stark contrast to WT cells which efficiently complete the RVD process. These findings corroborate earlier studies underscoring Piezo1 pivotal function in cell volume regulation across different cell types and experimental systems (Cahalan et al., 2015; Sforna et al., 2022). Our findings thus reinforce the concept that Piezo1 functions as a key mechanotransducer that couples mechanical stimuli to ion channel activation, thereby enabling cell volume recovery under osmotic stress.

The inability of Piezo1 KO cells to restore their volume following hypo-osmotic shock closely resembles the effects observed with pharmacological inhibition of $K_{Ca}$ channels (Michelucci et al., 2023), suggesting a mechanistic link between Piezo1 and $K_{Ca}$ channel activation in the context of volume regulation. This functional connection is likely to be mediated by $Ca^{2+}$, the common denominator between the two channel types, being the ion conducted by Piezo1 and the activator of $K_{Ca}$ channels. Accordingly, Piezo1-dependent $Ca^{2+}$ influx appears to serve as an essential upstream signal for $K_{Ca}$ channel opening during the RVD response, highlighting a coordinated interplay between these channels in maintaining cell volume homeostasis under osmotic stress.

Our electrophysiological data clearly demonstrate that Piezo1 is critical for the activation of both IK and BK channels during hypotonic stress, as evidenced by the complete absence of hypotonic-induced $K_{Ca}$ current

activation in Piezo1 KO cells. This lack of activation closely parallels the effects observed under conditions of external $Ca^{2+}$ removal or following application of the non-selective mechanosensitive channel inhibitor $Gd^{3+}$ (Michelucci et al., 2023). Together, these findings strongly support the conclusion that Piezo1-mediated $Ca^{2+}$ influx is an essential upstream event required for $K_{Ca}$ channel activation during RVD. Importantly, genetic ablation of Piezo1 does not affect the expression or functional capacity of $K_{Ca}$ channels. In fact, the maximal IK current evoked by the IK channel opener NS309 under

conditions of elevated intracellular $Ca^{2+}$, as well as the BK current recorded at depolarized voltages under isotonic conditions, are comparable between WT and Piezo1 KO cells. Notably, a modest hypotonic-induced activation of BK currents at strongly depolarized potentials ($>50$ mV) is still observed in Piezo1-deficient cells. Although this activation is absent at physiologically relevant voltages and contributes minimally to overall BK channel activity in the absence of Piezo1, it warrants consideration. One plausible explanation is a slight increase in intracellular $Ca^{2+}$ levels, potentially mediated by alternative

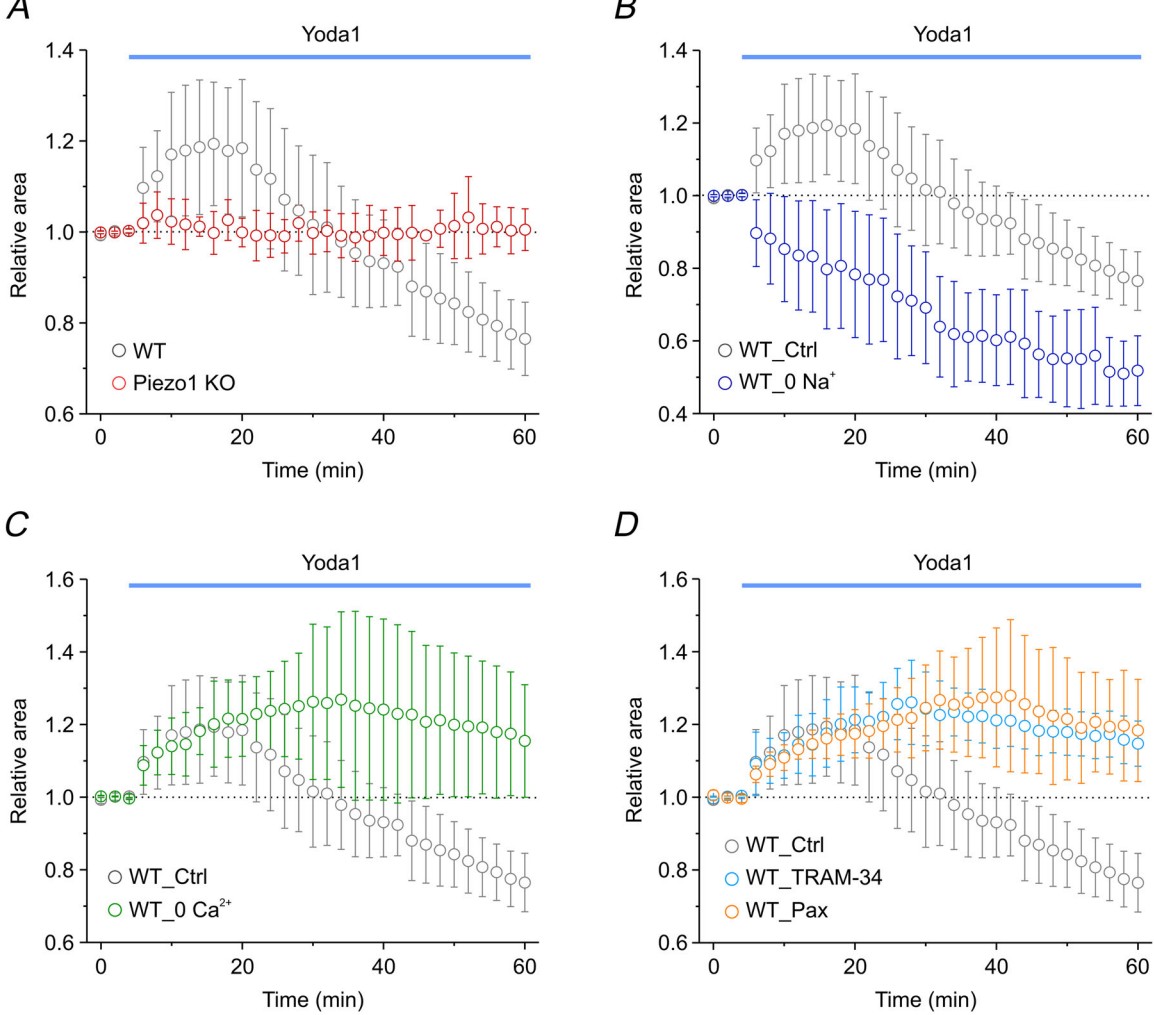

**Figure 8. Assessment of Yoda1-mediated volume changes in WT and Piezo1 KO U87-MG cells, using videoimaging analysis**

*A*, time course of relative cell area changes in isotonic conditions before (first three points) and during application of 10 μM Yoda1 (blue bar) in WT (grey circles, *n* = 7 cells from four independent cultures) and in Piezo1 KO (red circles, *n* = 7 cells from four independent cultures) cells. *B–D*, time courses of the relative cell area changes in WT cells during application of Yoda1 under different experimental conditions. The WT_Ctrl (grey circles) represents the reference group measured in standard extracellular Ringer solution and is identical to that shown in *A*. *B*, relative cell area changes in $Na^+$-free extracellular solution (dark blue circles, *n* = 6 cells from three independent cultures). *C*, relative cell area changes in $Ca^{2+}$-free extracellular solution (green circles, *n* = 6, from three independent cultures). *D*, relative cell area changes in the presence of either TRAM-34 (cyan circles, *n* = 5, from three independent cultures) or paxilline (orange circles, *n* = 5, from three independent cultures). Data are shown as mean ± SD.

$Ca^{2+}$-permeable mechanosensitive channels or by $Ca^{2+}$ release from intracellular stores such as the endoplasmic reticulum. Indeed, other mechanosensitive channels, including TRPV4, are expressed in GBM cells and have been reported to mediate hypotonic-induced $Ca^{2+}$ influx in related GBM lines (Lanciotti et al., 2012).

In light of this, the modest BK channel activation observed in Piezo1 KO cells may plausibly result from $Ca^{2+}$ entry through TRPV4 channels. However, this $Ca^{2+}$ influx appears insufficient to effectively activate BK channels at physiologically relevant membrane potentials and does not contribute meaningfully to volume regulation in U87-MG cells. This interpretation is further supported by recent evidence showing that TRPV4-mediated $Ca^{2+}$ influx is dispensable for hypotonic volume regulation in keratinocytes (Ritzmann et al., 2023), consistent with our present findings. Moreover, a direct activation of BK channels by cell swelling or osmotic stress can be excluded, as we previously demonstrated that no current activation occurs at any membrane potential when extracellular $Ca^{2+}$ influx is prevented (Michelucci et al., 2023).

In contrast to the $Ca^{2+}$-dependent activation of $K_{Ca}$ currents, VRAC-mediated $I_{Cl,swell}$ is unaffected by Piezo1 ablation, suggesting that VRAC activation occurs independently of Piezo1. This conclusion is further supported by our finding that pharmacological activation of Piezo1 with Yoda1 fails to elicit VRAC-mediated $I_{Cl,swell}$. These findings align with previous studies in GBM cell lines (Catacuzzeno et al., 2014; Sforna et al., 2017), which demonstrated that VRAC activation in response to hypotonic stress is $Ca^{2+}$ independent and proceeds trhough a mechanistically distinct pathway from that of $K_{Ca}$ channels. This stands in contrast to our earlier work on HEK293T cells, where Piezo1 was required for full $Ca^{2+}$-dependent activation of VRAC (Sforna et al., 2022), although the underlying mechanism remained undefined. One plausible explanation is that HEK293T cells may express $Ca^{2+}$-sensitive modulators that potentiate VRAC activity. Supporting this notion, our recent work showed that the astrocyte-specific MLC1 protein confers $Ca^{2+}$ sensitivity to an otherwise $Ca^{2+}$-insensitive VRAC (Brignone et al., 2022). An additional intriguing observation is that while Yoda1 promotes cell swelling, presumably through cation (i.e., $Na^+$) influx, it fails to activate VRAC. This reinfores the view that VRAC gating in GBM cells is not simply triggered by cell swelling *per se,* but also requires a reduction in intracellular ionic strength, as proposed in previous studies (Kasuya & Nureki, 2022; Osei-Owusu et al., 2018; Strange et al., 2019; Syeda et al., 2016). Since Piezo1 activation by Yoda1 promotes cation influx withouth a corresponding decrease in intracellular ionic

strength, the conditions necessary for VRAC activation are not met. Together, these findings highlight that VRAC activation in GBM cells relies on ionic strength-dependent mechanisms that are not engaged by Piezo1 activity alone.

Perhaps, the most unexpected finding of this study is that pharmacological activation of Piezo1 under isotonic conditions induces a biphasic volume response in WT U87-MG cells, consisting of an initial swelling followed by a marked shrinkage. The swelling phase is surprising, as Piezo1 activation by Yoda1 is generally thought to promote cell shrinkage via downstream activation of $K_{Ca}$ channels. However, given that Piezo1 is a non-selective cation channel permeable to $Na^+$, $K^+$ and $Ca^{2+}$, we propose that the early swelling phase reflects the influx of cations, primarily $Na^+$, due to its dominant concentration compared to the other permeable cations, accompanied by osmotically driven water entry, whereas the delayed shrinkage results from $Ca^{2+}$-dependent $K^+$ efflux through $K_{Ca}$ channels and the associated water loss. This interpretation is supported by dedicated experiments showing that the replacement of external $Na^+$ with impermeaant $NMDG^+$ abolished the swelling response, while removal of external $Ca^{2+}$ or pharmacologically blockade of IK and BK channels impaired the shrinkage phase. Overall, the biphasic volume response observed in WT cells underscores the multifaceted role of Piezo1 in volume regulation, reflecting its ability to mediate distinct ion fluxes and to activate $K_{Ca}$ channels in a $Ca^{2+}$-dependent manner. These findings demondtrate that Piezo1 can regulate cell volume not only in response to osmotic perturbations but also under isotonic conditions, potentially through as-yet-unidentified physiological modulators of channel activity. Moreover, the observation that Piezo1 activation can influence cell volume independently of $Ca^{2+}$-activated channels suggests a broader physiological relevance, potentially extending its function across diverse cell types and contexts beyond classical osmotic challenges.

A limitation of the present study lies in the use of projected cell area as a surrogate measure for cell volume. Although this approach is widely employed and enables real-time monitoring of volume dynamics in live cells, it does not account for the non-linear relationship between surface area and volume, particularly in cells with dynamic or non-spherical morphologies. Consequently, it lacks quantitative accuracy provided by direct volumetric techniques, such as calcein fluorescence quenching or three-dimensional *z*-stack reconstruction by confocal microscopy. Incorporating these complementary techniques in future studies would improve the resolution and reliability of volume measurements, thereby strengthening the conclusions drawn here.

## Piezo1 as a potential cell volume modifier in different cell types

Compelling early evidence for the crucial role of Piezo1 in volume regulation emerged from studies on gain-of-function mutations in the *PIEZO1* gene, which are associated with reduced erythrocyte volume (dehydration) and haemolytic anaemia, in disorders such as xerocytosis and dehydrated stomatocytosis (Albuisson et al., 2013; Bae et al., 2013). The molecular basis of this phenotype has been elucidated by two studies demonstrating that Piezo1-mediated $Ca^{2+}$ influx activates IK channel activity, promoting $K^+$ efflux and generating an osmotic gradient that drives water loss (Cahalan et al., 2015; Danielczok et al., 2017). This volume reduction facilitates erythrocyte deformation and efficient transit through narrow capillaries. More recently, we showed that *PIEZO1* deletion in HEK293T cells impairs RVD following osmotic swelling, whereas its overexpression accelerates volume recovery (Sforna et al., 2022). In the present study, we extend these findings to GBM cells, providing robust evidence that Piezo1 is the primary, if not exclusive, mechanosensitive channel responsible for mediating cell volume regulation in this context. Furthermore, we delineate the mechanistic link between Piezo1 and the activation of IK and BK channels, which parallels the pathway described in erythrocytes, where $Ca^{2+}$ influx through Piezo1 triggers IK channel activation and consequent water efflux (Cahalan et al., 2015; Danielczok et al., 2017).

It is important to acknowledge that Piezo1-mediated $Ca^{2+}$ influx and the subsequent activation of $Ca^{2+}$-activated channels are likely to be essential for RVD only in cell types where this process is $Ca^{2+}$ dependent, thus limiting the validity of this mechanism across different cell systems. However, the $Ca^{2+}$-dependent pathway involving $K_{Ca}$ channels is not the sole means by which Piezo1 controls cell volume. Our findings demonstrate that Piezo1 can also directly modify cell volume via $Na^+$ influx, which increases intracellular osmolarity and drives osmotically mediated water entry. Notably, such $Na^+$-dependent volume increases, referred to as *isotonic* or *isosmotic swelling*, can occur under both physiological and pathological conditions, even in the absence of extracellular hypotonicity. This $Na^+$-mediated mechanism suggests that Piezo1 may contribute to volume regulation in cell types that do not rely on $Ca^{2+}$ signalling. However, this possibility remains to be directly established and warrants further investigation.

Another important consideration is that cell volume regulation encompasses more than the classical RVD process. Deviations from the homeostatic volume set point can manifest as both transient and sustained increases in cell volume, including those associated with isotonic or isosmotic swelling. Our observation of $Na^+$-dependent volume changes highlights the potential of Piezo1 to participate in a broader spectrum of volume regulatory processes, including those elicited by mechanical stimuli or occurring under steady-state physiological conditions. Notably, such responses may operate independently of $Ca^{2+}$-dependent pathways, expanding the functional relevance of Piezo1 beyond its established role in RVD.

An important question that arises is whether the relatively small Piezo1 current elicited by Yoda1 under isotonic conditions is sufficient to account for the observed $\sim20\%$ increase in cell volume. At $-40$ mV, which closely approximates the resting membrane potential of U87-MG cells, the Yoda1-induced Piezo1 current density has been reported to be $\sim0.57$ pA/pF (Michelucci et al., 2023). Given a mean membrane capacitance of 35 pF, this corresponds to a total inward current of $\sim20$ pA. Assuming $Na^+$ is the predominant charge carrier under these conditions (Gnanasambandam et al., 2015), the estimated $Na^+$ influx is $\sim2.1\times10^{-16}$ mol/s. For a spherical cell with a diameter of $\sim20$ μm (volume $\approx$ 4.2 pL), this would result in a $Na^+$ concentration increase of approximately 50 μM/s. Over the 12 minutes required to reach peak swelling in WT U87-MG cells (Fig. 8), intracellular $Na^+$ levels could rise by $\sim36$ mM. If electroneutrality is preserved through parallel $Cl^-$ influx or retention, the total osmolarity increase would be $\sim72$ mOsm. Relative to an initial intracellular osmolarity of $\sim300$ mOsm, this corresponds to a $\sim24\%$ increase, consistent wite the approximately 20% increase in cell volume observed experimentally, assuming ideal osmotic behaviour ($\Delta V/V_0 \approx \Delta C/C_0$). These calculations support the conclusion that even small Piezo1-mediated $Na^+$ currents are sufficient to generate significant osmotic water influx. This provides quantitativesupport for a $Ca^{2+}$-independent mechanism by which Piezo1 can modulate cell volume under isotonic conditions. Future studies should explore this $Na^+$-dependent pathway in greater depth to fully elucidate Piezo1 role in maintaining volume homeostasis across different physiological and pathological contexts.

## Possible physio-pathological implications of Piezo1-mediated cell volume regulation in GBM

GBM is the deadliest form of human brain tumour, with a median patient survival of approximately 15 months (Davis, 2018; Davis et al., 1998). Its lethality is largely attributed to the highly invasive nature of GBM cells, which infiltrate the surrounding brain parenchyma, leading to tumour regrowth outside the central mass. This diffuse invasion severely limits the success of

surgical resection and worsens prognosis. Increasing evidence indicates that mechanical cues within the tumour microenvironment critically influence GBM behaviour, including migration and invasion. In this context, mechanosensitive channels, such as Piezo1, have emerged as key players. Piezo1 is reportedly overexpressed in GBM, with its expression inversely correlated with patient survival (Chen et al., 2018; Qu et al., 2020). However, the precise mechanismby which Piezo1 promotes GBM progression and malignancy remains incompletely understood.

To successfully infiltrate the narrow extracellular spaces of healthy brain tissue, GBM cells must dynamically adjust their volume and shape (Mcferrin & Sontheimer, 2006; Watkins & Sontheimer, 2011). This capacity for rapid morphological adaptation relies heavily on the activity of ion channels that regulate cell volume by driving osmotically mediated water fluxes. Among these, both IK and BK channels (Catacuzzeno et al., 2014; Sciaccaluga et al., 2010; Soroceanu et al., 1999; Turner et al., 2014), two of the most abundantly expressed $K^+$ channels in GBM cells, have been consistently linked to GBM cell migration and invasiveness. Similarly, Piezo1, which we have identified as a critical upstream regulator of $K_{Ca}$ channel activation during cell volume regulation, has been implicated in promoting GBM cell motility and invasiveness (Chen et al., 2018).

The findings from this study, along with our previous work (Michelucci et al., 2023), establish the Piezo1/$K_{Ca}$ axis as a key mechanism governing cell volume regulation in U87-MG cells. In erythrocytes, Piezo1-mediated activation of IK channel facilitates cell shrinkage, enabling passage through narrow capillaries (Cahalan et al., 2015). A similar mechanism may operate in GBM cells, where Piezo1-dependent activation of $K_{Ca}$ channels could permit dynamic shape adaptation during infiltration of the brain parenchyma. It is therefore plausible that mechanical stimuli within the tumour microenvironment, such as matrix stifness, induce local plasma membrane deformation, thereby activating Piezo1 and triggering $Ca^{2+}$ influx and downstream $K_{Ca}$ activation. This sequence of events would enable the volume and shape remodeling required for effective invasion. However, it is important to note that this Piezo1/$K_{Ca}$ channel axis has thus far been demonstrated only in the U87-MG cell line. Given the marked intra-tumoural heterogenity of GBM, caution is warranted in generalizing these findings . Further investigations in additional GBM cell lines and models, particularlypatient-derived primary cultures will be essential to determine the broader applicability and the clinical relevance of the Piezo1/$K_{Ca}$ signalling pathway in GBM pathophysiology.

## Conclusion

In summary, this study establishes the pivotal role of Piezo1 in the regulation of cell volume in a human GBM cell line. The principal mechanism by which Piezo1 exerts this function, under both hypotonic and isotonic conditions, involves the $Ca^{2+}$-dependent activation of the two major $K_{Ca}$ channels expressed in these cells. This mechanism may be particularly relevant to key malignant behaviours of GBM cells, including invasiveness and resistance to cell death. However, the downstream effects of Piezo1-mediated $Ca^{2+}$ influx likely extend beyond $K_{Ca}$ channel activation and may include regulation of cytoskeletal dynamics, additional ion channels, or volume-sensitive transporters. Further investigation into these additional signalling pathways will be necessary to fully elucidate the multifaceted physiological and pathological roles of Piezo1. Collectively, these findings pave the way for future studies exploring Piezo1 as a potential therapeutic target in GBM and other diseases where dysregulated volume control contributes to their progression.

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

# Additional information

## Data availability statement

All relevant information supporting the findings of this study are presented in the manuscript and in Supplementary Materials.

## Competing interests

The authors declare no conflict of interest.

## Author contributions

A.M. and L.C. conceived and designed the study. A.M., M.V.L., A.D.B., and L.S. conducted all patch clamp recordings and videoimaging experiments and analysed the data. L.D. and F.M.

performed western blot experiments and analysed the data. M.D.C. generated the Piezo1 KO cell model. A.M. and L.C. prepared the initial draft of the manuscript. All authors revised the manuscript critically for important intellectual content, approved the final version of the manuscript and agreed to be accountable for all aspects of the work. All persons designated as authors qualify for authorship, and all those who qualify for authorship are listed.

## Funding

Ministero dell'Università e della Ricerca (MUR): Luigi Catacuzzeno, 20223XZ5ER.

## Acknowledgements

This work has been supported by the following grants: Fondazione Cassa di Risparmio di Perugia (Project No. J95F21000050007); 'Kinetic models of ion channels: from atomic structures to membrane currents' funded by MUR PRIN 2022 (grant 20223XZ5ER, CUP J53D23006940006).

Open access publishing facilitated by Universita degli Studi di Perugia, as part of the Wiley - CRUI-CARE agreement.

## Keywords

glioblastoma, $K_{Ca}$ channels, Piezo1, RVD, volume regulation

## Supporting information

Additional supporting information can be found online in the Supporting Information section at the end of the HTML view of the article. Supporting information files available:

**Peer Review History**

