## [Peer Review History · The Journal of Physiology]

Piezo1 is an essential player in volume regulation of human glioblastoma cells

Antonio Michelucci, Maria Vittoria Leonardi, Angela Di Battista, Leonardo Donati, Luigi Sforza, Francesco Morena, Manlio Di Cristina, and Luigi Catacuzzeno

DOI: 10.1113/JP289215

Corresponding author(s): Antonio Michelucci (antonio.michelucci@unipg.it)

The following individual(s) involved in review of this submission have agreed to reveal their identity: León Islas (Referee #2); Sean Williams (Referee #3)

Review Timeline:

Submission Date:	10-May-2025
Editorial Decision:	10-Jun-2025
Revision Received:	27-Jun-2025
Editorial Decision:	16-Jul-2025
Revision Received:	20-Jul-2025
Accepted:	22-Jul-2025

Senior Editor: Peking Fong

Reviewing Editor: Pawel Ferdek

Transaction Report:

Dear Dr Michelucci,

Re: JP-RP-2025-289215 "Piezo1 is an essential player in volume regulation of human glioblastoma cells" by Antonio Michelucci, Maria Vittoria Leonardi, Angela Di Battista, Leonardo Donati, Luigi Sforza, Francesco Morena, Manlio Di Cristina, and Luigi Catacuzzeno

Thank you for submitting your manuscript to The Journal of Physiology. It has been assessed by a Reviewing Editor and by 2 expert referees and we are pleased to tell you that it is acceptable for publication following satisfactory revision.

REVISION CHECKLIST:

We look forward to receiving your revised submission.

Yours sincerely,

Peying Fong
Senior Editor
The Journal of Physiology

REQUIRED ITEMS

- Include a Key Points list in the article itself, before the Abstract.
- Author photo and profile. First or joint first authors are asked to provide a short biography (no more than 100 words for one author or 150 words in total for joint first authors) and a portrait photograph. These should be uploaded and clearly labelled together in a Word document with the revised version of the manuscript. See Information for Authors for further details.
- The contact information for the person responsible for 'Research Governance' at your institution needs to be provided. This includes their name and an institutional email address. Please ensure the contact is not an author on this paper and provide an alternate contact if necessary, or confirm in the submission form that the author whose email was provided has sole responsibility for research governance. This is the person who is responsible for regulations, principles and standards of good practice in research carried out at the institution, for instance the ethical treatment of animals, the keeping of proper experimental records or the reporting of results.
- You must start the Methods section with a paragraph headed Ethical Approval. If experiments were conducted on humans, confirmation that informed consent was obtained, preferably in writing, that the studies conformed to the standards set by the latest revision of the Declaration of Helsinki and that the procedures were approved by a properly constituted ethics committee, which should be named, must be included in the article file. If the research study was registered (clause 35 of the Declaration of Helsinki), the registration database should be indicated, otherwise the lack of registration should be noted as an exception (e.g. The study conformed to the standards set by the Declaration of Helsinki, except for registration in a database). For further information see: <https://physoc.onlinelibrary.wiley.com/hub/human-experiments>.
- Your manuscript must include a complete Additional Information section, including competing interests; funding; author contributions and acknowledgements.
- The Journal of Physiology funds authors of provisionally accepted papers to use the premium BioRender site to create high resolution schematic figures. Follow this link and enter your details and the manuscript number to create and download figures. Upload these as the figure files for your revised submission. If you choose not to take up this offer, we require figures to be of similar quality and resolution. If you are opting out of this service to authors, state this in the Comments section on the Detailed Information page of the submission form. The link provided should only be used for the purposes of this submission. Authors will be charged for figures created on this premium BioRender account if they are not related to this manuscript submission.
- Please upload separate high-quality figure files via the submission form.
- You must upload original, uncropped western blot/gel images (including controls) if they are not included in the manuscript. This is to confirm that no inappropriate, unethical or misleading image manipulation has occurred. These should be uploaded as 'Supporting information for review process only'. Please label/highlight the original gels so that we can clearly see which sections/lanes have been used in the manuscript figures. For more information, see: <https://physoc.onlinelibrary.wiley.com/hub/journal-policies#magmanip>.
- Papers must comply with the Statistics Policy: https://jp.msubmit.net/cgi-bin/main.plex?form_type=display_requirements#statistics.

In summary:

- If n {less than or equal to} 30, all data points must be plotted in the figure in a way that reveals their range and distribution. A bar graph with data points overlaid, a box and whisker plot or a violin plot (preferably with data points included) are acceptable formats.

- If $n > 30$, then the entire raw dataset must be made available either as supporting information, or hosted on a not-for-profit repository, e.g. FigShare, with access details provided in the manuscript.
- 'n' clearly defined (e.g. x cells from y slices in z animals) in the Methods. Authors should be mindful of pseudoreplication.
- All relevant 'n' values must be clearly stated in the main text, figures and tables.
- The most appropriate summary statistic (e.g. mean or median and standard deviation) must be used. Standard Error of the Mean (SEM) alone is not permitted.
- Exact p values must be stated. Authors must not use 'greater than' or 'less than'. Exact p values must be stated to three significant figures even when 'no statistical significance' is claimed.
- Please include an Abstract Figure file, as well as the Figure Legend text within the main article file. The Abstract Figure is a piece of artwork designed to give readers an immediate understanding of the research and should summarise the main conclusions. If possible, the image should be easily 'readable' from left to right or top to bottom. It should show the physiological relevance of the manuscript so readers can assess the importance and content of its findings. Abstract Figures should not merely recapitulate other figures in the manuscript. Please try to keep the diagram as simple as possible and without superfluous information that may distract from the main conclusion(s). Abstract Figures must be provided by authors no later than the revised manuscript stage and should be uploaded as a separate file during online submission labelled as File Type 'Abstract Figure'. Please also ensure that you include the figure legend in the main article file. All Abstract Figures should be created using BioRender. Authors should use The Journal's premium BioRender account to export high-resolution images. Details on how to use and access the premium account are included as part of this email.
- Please include a full title page as part of your main article (Word) file, which should contain the following: title, authors, affiliations, corresponding author name and contact details, keywords, and running title.
- Please ensure that all figures and tables have a title and legend, and that they have been cited within the main article text.

EDITOR COMMENTS

Reviewing Editor:

Thank you for submitting your manuscript to The Journal of Physiology. This is an interesting and well-written study that provides valuable new insight into the role of Piezo1. Your work has now been reviewed by two independent Referees, who are enthusiastic about the scientific content and significance of your findings. However, both Referees have identified several areas that require your attention and possible revision.

Please note that The Journal of Physiology does not impose a formal word limit and does not normally accept supplementary material. You are therefore encouraged to include all essential content directly in the main manuscript, either in the main text or, if necessary, in an Appendix.

Please indicate how data normality was assessed, in order to justify the use of parametric or non-parametric statistical tests.

To comply with The Journal of Physiology's requirements, exact p-values (to three significant figures) must be reported in the main text, figures, legends, and tables. Asterisks alone are not permitted, and p-values < 0.001 may be reported as '<0.001'.

Senior Editor:

Review of your manuscript, "Piezo1 is an essential player in volume regulation of human glioblastoma cells" is now complete. Attached herewith for your consideration are critiques from two Expert Referees. Although both Expert Referees concur on the potential impact of your study, they do also outline in detail specific suggestions for improvement. Their queries encompass matters of over-generalizations in interpretation (Referee 1), to requests for improved transparency in presenting methodological details required for ensuring rigor (both Referees). Please use these to guide the revision of your manuscript.

Revised manuscripts are expected to comply with published journal policies, as highlighted in the Reviewing Editor's comments. I commend your attention to those pertaining to presentation of statistics, as well as the policy regarding supporting information.

Thank you for submitting your manuscript for The Journal of Physiology's consideration. We look forward to receiving your revised manuscript.

Statistics note:

The Authors are reminded to comply with The Journal of Physiology's Statistics Policy; elements are highlighted in the Reviewing Editor's Comments to the Authors.

REFeree COMMENTS

Referee #1:

This manuscript provides strong evidence for the critical role of the mechanosensitive cation/Ca²⁺ channel Piezo1 in regulating cell volume in the glioblastoma U87-MG cell line. The authors address this by comparing U87-MG cells that express Piezo1 with those in which Piezo1 expression has been ablated using CRISPR/Cas9. They employ straightforward microscopy-based measurements of cell volume under isoosmotic and hypoosmotic conditions. Additionally, they combine electrophysiology and pharmacological agents to demonstrate that Piezo1 activity is essential for the hypoosmotic activation of the Ca²⁺-sensitive K⁺ channels from the BK and IK sub-families.

The main findings of the manuscript are:

- In U87-MG cells, Piezo1 activity is critical for regulatory volume decrease (RVD), as Piezo1 deletion reduces the RVD rate by at least three-fold.
- Piezo1 activity is important for hypoosmotic activation of BK and IK channels in U87-MG cells, especially at physiologically relevant membrane potentials, with Piezo1 deletion significantly impairing hypoosmotic BK and IK activation.
- Pharmacological activation of Piezo1 with Yoda1 in U87-MG cells triggers biphasic changes in cell volume: an initial transient swelling due to Na⁺ influx, followed by sustained shrinkage due to K⁺ loss through BK and IK channels..

Below are several suggestions for improving manuscript clarity:

Major-to-moderate concerns:

[1] The authors may overstate the generality of Piezo1's role in RVD across cell types. They hint that Piezo1 is broadly important for RVD, but this requires careful evaluation. In many cell types, RVD does not depend (or depends only weakly) on extracellular Ca²⁺ levels, and thus is unlikely to involve plasmalemmal Ca²⁺ permeability (as reviewed by McCarthy and O'Neil, *Physiol. Rev.*, 1992; Hoffmann et al., *Physiol. Rev.*, 2009). Therefore, I suggest removing generalized headings such as "Piezo1 as a general cell volume regulator in different cell types" (line 598). Currently, Piezo1's role in cell volume regulation and/or homeostasis has been clearly demonstrated only in erythrocytes (evidence from multiple groups), HEK293 cells (prior work by this group), and U87-MG cells (the present study).

[2] A similar concern applies to the manuscript's discussion of Piezo1's significance in glioblastoma (GBM). GBM is known for its marked inter-tumoral heterogeneity, which is reflected in the outdated term "glioblastoma multiforme." This study focuses solely on the U87-MG cell line (derived from the original Uppsala 87 Malignant Glioma stock). U87-MG cells have unique features compared to other GBM cell lines, including morphology, substrate adhesion, and signaling pathways. While they carry the typical PTEN, PI3K, and Akt mutations, they also have a rare homozygous deletion of p16. Although the literature on Piezo1 in GBM malignancy is growing, this does not imply a universal role in GBM cell volume regulation. It would strengthen the manuscript to include a clear disclaimer that these findings should be replicated in multiple primary GBM cultures to confirm their broader relevance.

[3] In Figure 6, the authors use the same set of controls (n=6) for comparisons with four independent treatment conditions. This should be explicitly disclosed in both the main text and the figure legend. While this reviewer does not consider this a major flaw affecting the overall conclusions, it is important to acknowledge that comparisons should ideally be made to contemporary controls in the same cell preparations.

[4] Lns. 571-575. The finding that Yoda1-induced cell swelling does not trigger VRAC activation is intriguing. However, the authors' explanation that VRAC is activated by reductions in ionic strength rather than cell swelling is consistent with the bulk of evidence accumulated by the field. Strange and colleagues discuss this distinction in detail in their 2019 review in *J. Gen. Physiol.* The manuscript would benefit from reconsidering this explanation and aligning it more closely with the current understanding of VRAC activation.

Minor concerns:

[5] Several references in this manuscript lack full bibliographic information (including volume and page numbers). Please update the following references: Albuissou et al., 2013; Bae et al., 2013; Brignone et al., 2022; Cahalan et al., 2015; Danielczok et al., 2017; Dubinsky et al., 2000; Felipe et al., 1993; Guerriero et al., 2024; Kasuya and Nureki, 2022; Klumpp et al., 2018; Sciacaluga et al., 2010; Serpe et al., 2022; Syeda et al., 2015; Wang et al., 2003.

[6] Lns. 89-90 and 332-333: The statement regarding the lack of selective inhibitors for Piezo1 may be overstated. The

Grammostola mechanotoxin GsMTx4 is widely used as a selective pharmacological inhibitor of Piezo1 and should be acknowledged as such.

[7] Lns. 187-190. The authors mention that RVD was measured in the same solutions as used in electrophysiological experiments. It would be important to clarify that octanol, which is known to partially block VRACs and thus interfere with RVD, was not included in the RVD measurements.

[8] It is rather surprising that cell volume measurements were performed in so few cells. Given that 40x magnification imaging fields typically contain multiple cells, it would be useful to clarify how many independent RVD experiments were conducted. This is a key point that impacts evaluation of the rigor and reproducibility of the results.

[9] In the calculation of cell volume, the authors use changes in cell area as a proxy for volume changes. While this approach may capture qualitative trends, it is quantitatively inaccurate because the cross-sectional area ($A = \pi r^2$) and cell volume ($V = 4/3 \pi r^3$) scale differently. This should be acknowledged as a limitation, although the overall conclusions appear to remain valid.

Referee #2:

The manuscript by Michelucci et al. describes a study of the role of Piezo1 channels in cell volume regulation in a glioblastoma cell line. The authors had previously shown that Piezo1 mediates volume-induced cation entry and this in turn activates potassium efflux mediated by calcium-activated BK channels. In the present manuscript, the authors have used CRISPR-cas9 based approaches to generate a cell line with Piezo1 knocked-out. This has allowed them to better delineate the essential role of Piezo 1 in cell volume regulation in these cells.

The experiments are carefully carried out and interpreted and the results are clear. This article is a fine contribution to help understand the role of Piezo channels in sensing mechanical changes in the membrane during volume regulation.

1. Please clarify if the knock-out of Piezo1 eliminates the whole protein or produces a truncated protein. This is important since some deletions of Piezo1 are not functional but might still influence the mechanical response of the membrane. Also, the lack of detection of protein in the western blot could be due to the absence of the epitope if a truncated protein is produced.

2. Figure 1 seems to indicate that there is no mechanical response due to Yoda1 in deltaP5 cells. However, it would be surprising that no other currents can be mechanically activated. Did you try activating with a mechanical stimulus and not only Yoda1?

3. In figure 5D, if this is a WT cell, why yoda1 application did not activate a Piezo1 mediated current as in Figure 1B? Are these recordings obtained in the absence of cations?

4. In the discussion section, the small activation of BK channels is attributed to a possible small increase of intracellular Ca^{2+} through other mechanically gated channels. Can you indicate what other mechanically gated channels are present in these cells? And if they are known to mediate Ca^{2+} permeation?

5. The current density produced by Piezo1 activation by Yoda1 is small (Figure 1), is this flux of Na^{+} and Ca^{2+} expected to produce the ~20% increase in volume in isotonic conditions (Figure 6)?

END OF COMMENTS

EDITOR COMMENTS

Reviewing Editor:

Thank you for submitting your manuscript to The Journal of Physiology. This is an interesting and well-written study that provides valuable new insight into the role of Piezo1. Your work has now been reviewed by two independent Referees, who are enthusiastic about the scientific content and significance of your findings. However, both Referees have identified several areas that require your attention and possible revision.

We would like to sincerely thank the Reviewing Editor for his/her positive and encouraging comments regarding our manuscript. We are grateful that you acknowledged the importance of our work and recognized the overall quality of the manuscript. We appreciate the thoughtful feedback from both expert Referees, and we have carefully considered all of the points raised and have revised the manuscript accordingly. A detailed response to each of the Reviewers' comments is provided below. We hope that the changes we have made address the concerns raised and further strengthen the manuscript.

Please note that The Journal of Physiology does not impose a formal word limit and does not normally accept supplementary material. You are therefore encouraged to include all essential content directly in the main manuscript, either in the main text or, if necessary, in an Appendix.

All figures and results have now been incorporated into the main manuscript, and the Supplementary Material section just contains the original unprocessed/uncropped Western blot gels.

Please indicate how data normality was assessed, in order to justify the use of parametric or non-parametric statistical tests.

Data normality was assessed using the Shapiro–Wilk test to determine the appropriate use of parametric or non-parametric statistical analyses.

To comply with The Journal of Physiology's requirements, exact p-values (to three significant figures) must be reported in the main text, figures, legends, and tables. Asterisks alone are not permitted, and p-values < 0.001 may be reported as '<0.001'.

We have reported all exact p-values (to three significant figures) in the figures and have explicitly stated the statistical tests used throughout the manuscript.

Senior Editor:

Review of your manuscript, "Piezo1 is an essential player in volume regulation of human glioblastoma cells" is now complete. Attached herewith for your consideration are critiques from two Expert Referees. Although both Expert Referees concur on the potential impact of your study, they do also outline in detail specific suggestions for improvement. Their queries encompass matters of over-generalizations in interpretation (Referee 1), to requests for improved transparency in presenting methodological details required for ensuring rigor (both Referees). Please use these to guide the revision of your manuscript.

We sincerely thank the Senior Editor for his/her thoughtful comments and for overseeing the review of our manuscript. We are grateful to both expert Referees for their constructive feedback and for recognizing the potential impact of our study. We have carefully considered all suggestions and concerns raised, particularly those related to over-generalizations in our interpretations and the need for enhanced methodological transparency. In response, we have revised the manuscript to ensure a more balanced and evidence-based discussion of our findings, and we have expanded the Methods section to provide greater clarity and detail. These changes aim to improve the overall rigor and transparency of the work in line with the journal's standards.

Below, we provide a point-by-point response to each of the Referees' comments, indicating how each concern has been addressed in the revised manuscript. We hope that these revisions meet your expectations and those of the reviewers.

Revised manuscripts are expected to comply with published journal policies, as highlighted in the Reviewing Editor's comments. I commend your attention to those pertaining to presentation of statistics, as well as the policy regarding supporting information.

We have carefully addressed all points raised, including those related to statistical reporting and the presentation of supporting information. All statistical results are now reported in accordance with the Journal's guidelines, and all essential content has been incorporated directly into the main manuscript, with the supplementary material section just containing the unprocessed immunoblots.

Thank you for submitting your manuscript for The Journal of Physiology's consideration. We look forward to receiving your revised manuscript.

Statistics note:

The Authors are reminded to comply with The Journal of Physiology's Statistics Policy; elements are highlighted in the Reviewing Editor's Comments to the Authors.

We have carefully reviewed and addressed all elements of The Journal of Physiology's Statistics Policy, as highlighted in the Reviewing Editor's comments. The manuscript has been revised accordingly to ensure full compliance with the journal's statistical reporting requirements.

REFEREE COMMENTS

Referee #1:

This manuscript provides strong evidence for the critical role of the mechanosensitive cation/ Ca^{2+} channel Piezo1 in regulating cell volume in the glioblastoma U87-MG cell line. The authors address this by comparing U87-MG cells that express Piezo1 with those in which Piezo1 expression has been ablated using CRISPR/Cas9. They employ straightforward microscopy-based measurements of cell volume under isoosmotic and hypoosmotic conditions. Additionally, they combine electrophysiology and pharmacological agents to demonstrate that Piezo1 activity is essential for the hypoosmotic activation of the Ca^{2+} -sensitive K^+ channels from the BK and IK sub-families.

The main findings of the manuscript are:

- In U87-MG cells, Piezo1 activity is critical for regulatory volume decrease (RVD), as Piezo1 deletion reduces the RVD rate by at least three-fold.
- Piezo1 activity is important for hypoosmotic activation of BK and IK channels in U87-MG cells, especially at physiologically relevant membrane potentials, with Piezo1 deletion significantly impairing hypoosmotic BK and IK activation.
- Pharmacological activation of Piezo1 with Yoda1 in U87-MG cells triggers biphasic changes in cell volume: an initial transient swelling due to Na^+ influx, followed by sustained shrinkage due to K^+ loss through BK and IK channels..

Below are several suggestions for improving manuscript clarity:

Major-to-moderate concerns:

[1] The authors may overstate the generality of Piezo1's role in RVD across cell types. They hint that Piezo1 is broadly important for RVD, but this requires careful evaluation. In many cell types, RVD does not depend (or depends only weakly) on extracellular Ca^{2+} levels, and thus is unlikely to involve plasmalemmal Ca^{2+} permeability (as reviewed by McCarthy and O'Neil, *Physiol. Rev.*, 1992; Hoffmann et al., *Physiol. Rev.*, 2009). Therefore, I suggest removing generalized headings such as "Piezo1 as a general cell volume regulator in different cell types" (line 598). Currently, Piezo1's role in cell volume regulation and/or homeostasis has been clearly demonstrated only in erythrocytes (evidence from multiple groups), HEK293 cells (prior work by this group), and U87-MG cells (the present study).

We sincerely thank the Reviewer for the thoughtful comment and for highlighting the variability in the Ca^{2+} dependency of RVD across different cell types. We agree that Piezo1-mediated Ca^{2+} influx is essential for RVD only in cells where this process is Ca^{2+} -dependent.

That said, we respectfully propose that Piezo1 may still play a significant role in cell volume regulation or modulation, even in cell types where Ca^{2+} is not required for RVD. This perspective is supported by the following considerations:

1 - Piezo1 also conducts Na^+ ions, and our results indicate that its Na^+ permeability can promote water influx and lead to cell swelling under isotonic conditions, independent of Ca^{2+} signaling (as shown in Figure 8B). Under both physiological and pathological conditions, cells can undergo volume increases due to elevated salt influx, a process termed *isotonic* or *isosmotic swelling*, which occurs even when the extracellular environment is not hypotonic. Our data suggest that Piezo1 may contribute to this form of swelling through a Ca^{2+} -independent pathway.

2 - Cell volume regulation extends beyond RVD. Our intention was not to suggest that Piezo1 universally mediates RVD across all cell types. Rather, we aimed to underscore that Piezo1 may influence a broader range of volume changes, including volume increases or transient swelling triggered by mechanical stimuli or occurring under baseline physiological conditions.

To address the Reviewer's concern, we have made the following revisions:

- Updated section headings to avoid implying that Piezo1 serves as a universal mediator of RVD, as follow: "*Piezo1 as a potential cell volume modifier in different cell types.*" (page 21, line 686).
- Clarified that Piezo1 contribution to RVD is limited to Ca^{2+} -sensitive cells, while its broader role in cell volume modulation may involve Na^+ influx and associated water movement (pages 21-22, lines 703-714).
- Explicitly distinguished between mechanisms that decrease cell volume (such as RVD) and those that lead to volume increases, emphasizing that both fall under the broader concept of volume regulation (page 22, lines 715-721)

We hope these revisions satisfactorily address the concern and enhance the clarity and balance of our discussion regarding Piezo1's role in cell volume regulation.

[2] A similar concern applies to the manuscript's discussion of Piezo1's significance in glioblastoma (GBM). GBM is known for its marked inter-tumoral heterogeneity, which is reflected in the outdated term "glioblastoma multiforme." This study focuses solely on the U87-MG cell line (derived from the original Uppsala 87 Malignant Glioma stock). U87-MG cells have unique features compared to other GBM cell lines, including morphology, substrate adhesion, and signaling pathways. While they carry the typical PTEN, PI3K, and Akt mutations, they also have a rare homozygous deletion of p16. Although the literature on Piezo1 in GBM malignancy is growing, this does not imply a universal role in GBM cell volume regulation. It would strengthen the manuscript to include a clear disclaimer that these findings should be replicated in multiple primary GBM cultures to confirm their broader relevance.

We thank the Reviewer for this well-founded observation regarding the heterogeneity of GBM and the limitations of using only the U87-MG cell line. We fully agree that GBM is a highly heterogeneous disease, and that findings derived from a single cell line such as U87-MG, while informative, cannot be generalized across all GBM subtypes or patient-derived tumors. We have toned down any language that may have implied a generalized

relevance of this pathway in GBM malignancy, and we now explicitly acknowledge the need for replication of our findings in diverse GBM models to confirm their broader applicability.

In response to this comment, we have revised the manuscript to more clearly state that (page 23, lines 770-776):

“However, this Piezo1/KCa channel axis, at present, has only been demonstrated in the U87-MG cell line. While the current findings are compelling, they should be considered preliminary within the broader context of GBM and interpreted with caution given the substantial inter-tumoral heterogeneity of GBM. Further investigations in additional GBM cell lines and, critically, in patient-derived primary cultures are needed to determine the broader applicability of this mechanism.”

We believe these changes improve the scientific rigor and transparency of the manuscript and we are grateful to the reviewer for prompting this clarification.

[3] In Figure 6, the authors use the same set of controls (n=6) for comparisons with four independent treatment conditions. This should be explicitly disclosed in both the main text and the figure legend. While this reviewer does not consider this a major flaw affecting the overall conclusions, it is important to acknowledge that comparisons should ideally be made to contemporary controls in the same cell preparations.

We thank the Reviewer for raising this point. We would like to clarify that all experiments were conducted contemporaneously. The WT cells used as controls for the Piezo1 KO group are the same WT cells used for comparison with the various experimental conditions (e.g., 0Ca, 0Na, TRAM-34, and paxilline), all performed within the same experimental sessions. On each experimental day and for every condition tested, at least one WT control was recorded in parallel, ensuring that all comparisons between controls and treatment groups were made under identical conditions.

Although six control cells may appear to be a relatively small number for comparison across all conditions, it is important to note that additional cells were excluded from the analysis due to the loss of spherical morphology, which prevented accurate estimation of relative cell area. It should also be noticed that typically less than 5 cells were present in the field in our experiments (using a 40x magnification and with 30000 cells plated on a 35 mm Petri dish).

This has now been explicitly clarified in the revised manuscript, where we have also clearly stated the number of replicates used (legend to Figure 8 and pages 6-7, lines 222-230).

[4] Lns. 571-575. The finding that Yoda1-induced cell swelling does not trigger VRAC activation is intriguing. However, the authors' explanation that VRAC is activated by reductions in ionic strength rather than cell swelling is consistent with the bulk of evidence accumulated by the field. Strange and colleagues discuss this distinction in detail in their 2019 review in J. Gen. Physiol. The manuscript would benefit from reconsidering this explanation and aligning it more closely with the current understanding of VRAC activation. We fully agree that the lack of VRAC activation by Yoda1 under isotonic conditions, despite the observed cell swelling, supports the idea that a reduction in intracellular ionic strength might be necessary to sensitize the VRAC channel to the swelling-induced

activation. This interpretation is indeed consistent with the current consensus in the field, as comprehensively discussed by Strange and colleagues (J. Gen. Physiol., 2019). In line with the Reviewer's suggestion, we have revised the relevant section of the manuscript to better reflect this distinction and to emphasize our view that ionic strength serves as a trigger for VRAC activation. This, in turn, may explain why Piezo1-mediated swelling alone is insufficient to activate VRAC. (page 20, lines 650-658): *"This discrepancy may be explained by prior studies suggesting that both a reduction in intracellular ionic strength and an increase in cell volume contribute to VRAC activation during hypotonic-induced cell swelling (Syeda et al., 2016; Osei-Owusu et al., 2018; Strange et al., 2019; Kasuya & Nureki, 2022). Notably, the cell swelling induced by Piezo1 activation, such as that elicited by Yoda1, is primarily driven by cation influx and is not expected to produce a concomitant reduction in ionic strength. This may account for the failure of VRAC activation under these conditions, highlighting the requirement for ionic strength changes in the gating of VRAC in GBM cells."*

We believe this clarification strengthens the mechanistic interpretation of our findings and aligns the manuscript more closely with the prevailing understanding in literature.

Minor concerns:

[5] Several references in this manuscript lack full bibliographic information (including volume and page numbers). Please update the following references: Albuissou et al., 2013; Bae et al., 2013; Brignone et al., 2022; Cahalan et al., 2015; Danielczok et al., 2017; Dubinsky et al., 2000; Felipe et al., 1993; Guerriero et al., 2024; Kasuya and Nureki, 2022; Klumpp et al., 2018; Sciacaluga et al., 2010; Serpe et al., 2022; Syeda et al., 2015; Wang et al., 2003.

We thank the Reviewer for catching these errors. We now fixed the indicated references.

[6] Lns. 89-90 and 332-333: The statement regarding the lack of selective inhibitors for Piezo1 may be overstated. The Grammostola mechanotoxin GsMTx4 is widely used as a selective pharmacological inhibitor of Piezo1 and should be acknowledged as such.

We thank the Reviewer for this important comment and fully acknowledge that GsMTx4 is a widely used inhibitor of Piezo1. However, we respectfully note that its selectivity is limited, as GsMTx4 has also been reported to modulate other mechanosensitive cation channels, including members of the TRP channel family (Spasova et al., 2006; Bowman et al., 2007; Alessandri-Haber et al., 2009).

In response to the Reviewer's suggestion, we have revised the relevant statements in the manuscript (page 3, lines. 90-99) to present a more balanced and accurate perspective.

"However, the lack of highly selective pharmacological inhibitors for Piezo1 has hindered definitive characterization of its functional roles. Although the well-characterized peptide GsMTx4 effectively inhibits Piezo1 activity, its limited specificity is a concern, as it also modulates other mechanosensitive cation channels, including members of the TRP family (Spasova et al., 2006; Bowman et al., 2007; Alessandri-Haber et al., 2009). Thus, while GsMTx4 remains a valuable a useful tool for probing mechanosensitive channel function, potential off-target effects demand cautious interpretation of results. In this context, genetic

approaches are essential for conclusively linking Piezo1 activity to specific cellular responses, and for excluding contributions from alternative mechanotransduction pathways.”

[7] Lns. 187-190. The authors mention that RVD was measured in the same solutions as used in electrophysiological experiments. It would be important to clarify that octanol, which is known to partially block VRACs and thus interfere with RVD, was not included in the RVD measurements.

Octanol was not included in the external solution during RVD measurements. We have now clarified this point in the main text to prevent any misunderstanding (page 6, lines 207-208).

[8] It is rather surprising that cell volume measurements were performed in so few cells. Given that 40x magnification imaging fields typically contain multiple cells, it would be useful to clarify how many independent RVD experiments were conducted. This is a key point that impacts evaluation of the rigor and reproducibility of the results.

We agree with the Reviewer’s comment. First, we note that not all cells within the imaging field were analyzed, as some lost their spherical shape during the experiment, making accurate measurement of relative area unreliable. Second, we have now specified the number of independent RVD experiments performed in the revised manuscript to clarify the rigor and reproducibility of our analysis (pages 6-7, lines 222-230)

[9] In the calculation of cell volume, the authors use changes in cell area as a proxy for volume changes. While this approach may capture qualitative trends, it is quantitatively inaccurate because the cross-sectional area ($A = \pi r^2$) and cell volume ($V = 4/3 \pi r^3$) scale differently. This should be acknowledged as a limitation, although the overall conclusions appear to remain valid.

We agree that using relative projection area as a proxy for cell volume is less precise than direct volumetric measurements obtained through methods such as calcein fluorescence or confocal z-stacks. In response, we have added a sentence to the manuscript acknowledging this quantitative limitation (page 21, lines 678-685). “A limitation of the present study is the use of projected cell area as a surrogate measure for cell volume. Although this approach is widely employed and enables qualitative assessment of volume dynamics in live cells, it does not account for the nonlinear relationship between surface area and volume, particularly in cells with irregular or dynamic morphologies. As such, it lacks quantitative accuracy provided by direct volumetric techniques, such as calcein fluorescence quenching or three-dimensional confocal z-stack reconstruction. Future studies incorporating these complementary methods would enhance the precision of volume measurements and further validate the findings reported here.”

Referee #2:

The manuscript by Michelucci et al. describes a study of the role of Piezo1 channels in cell volume regulation in a glioblastoma cell line. The authors had previously shown that Piezo1 mediates volume-induced cation entry and this in turn activates potassium efflux mediated by calcium-activated BK channels. In the present manuscript, the authors have used CRISPR-cas9 based approaches to generate a cell line with Piezo1 knocked-out. This has allowed them to better delineate the essential role of Piezo 1 in cell volume regulation in these cells.

The experiments are carefully carried out and interpreted and the results are clear. This article is a fine contribution to help understand the role of Piezo channels in sensing mechanical changes in the membrane during volume regulation.

1. Please clarify if the knock-out of Piezo1 eliminates the whole protein or produces a truncated protein. This is important since some deletions of Piezo1 are not functional but might still influence the mechanical response of the membrane. Also, the lack of detection of protein in the western blot could be due to the absence of the epitope if a truncated protein is produced.

As shown in Fig. 1, the 31 bp deletion in the *PIEZO1* gene introduces a frameshift after amino acid 56, resulting in a truncated protein composed of the first 56 wild-type residues followed by 38 unrelated amino acids before premature stop codon. This truncated product would retain only the initial transmembrane domain and is therefore incapable of assembling into a functional channel. Moreover, given its minimal size (just 56 of the ~2500 amino acids comprising the full-length Piezo1 protein) it is highly unlikely that this truncated form is stable. Such an aberrant and incomplete protein would be expected to undergo rapid degradation via cellular quality control mechanisms. Consequently, no significant levels of the truncated protein should be present.

The polyclonal antibody used in the Western blot recognizes the sequence corresponding to amino acids 2230–2420 of human PIEZO1, which is located near the C-terminus of the protein. If an alternative AUG downstream of the stop codon generated by the frameshift introduced by the deletion had been used, the resulting truncated form would still have been detected by our antibody. Therefore, the absence of signal in KO cells further supports the conclusion that Piezo1 protein is not expressed.

This clarification has now been added to the main text (Methods, page 4, lines 147-154) (Results, page 8, lines 273-285) as suggested.

2. Figure 1 seems to indicate that there is no mechanical response due to Yoda1 in deltaP5 cells. However, it would be surprising that no other currents can be mechanically activated. Did you try activating with a mechanical stimulus and not only Yoda1?

We thank the reviewer for this insightful comment. The primary aim of the experiments shown in Figure 1B-C (now Figure 2B-C in the revised manuscript) was to functionally validate the Piezo1 KO model generated via CRISPR/Cas9. To this end, we employed Yoda1, a small-molecule agonist that selectively activates Piezo1, which activates the

channel by directly binding to and stabilizing its open conformation, thus allowing us to assess Piezo1-specific activity independently of mechanical force.

We agree that mechanical stimulation can activate multiple mechanosensitive ion channels, including members of the TRP channel family, and that it would be unlikely for $\Delta P5$ cells to be entirely devoid of mechanically responsive currents. However, our experimental design focused on isolating Piezo1-mediated responses, and therefore we prioritized the use of Yoda1 to avoid confounding activation of non-Piezo1 channels. That said, we acknowledge that we did not apply mechanical stimuli (aside from hypotonic swelling) under conditions in which IK, BK, and VRAC currents were blocked in Piezo1 KO cells. Thus, we cannot exclude the presence of other minor mechanosensitive currents in these cells. As we discuss further in response to Point 4, channels such as TRPV4, which are expressed in GBM cells, may be mechanically activated and contribute to Ca^{2+} signaling, although current evidence suggests their role in volume regulation is limited under our conditions.

3. In figure 5D, if this is a WT cell, why yoda1 application did not activate a Piezo1 mediated current as in Figure 1B? Are these recordings obtained in the absence of cations?

The recordings shown in Figure 5D (now Figure 7D in the revised manuscript) were performed in standard Ringer solution containing all major cations, including Na^+ and Ca^{2+} which, at least in principle, permits the assessment of Piezo1 activity. The solution also contained TRAM-34 and paxilline to block K_{Ca} channel activity but, notably, did not include DCPIB, the VRAC inhibitor used in Figure 1B to isolate Piezo1-mediated currents. Under these conditions, we typically observe a basal $I_{Cl,swell}$ current already active in isotonic solution, with amplitudes in the range of 100–200 pA at -80 mV. Given an average membrane capacitance of ~ 35 pF, this corresponds to a current density of approximately 5 pA/pF, comparable to the Yoda1-activated Piezo1 current observed in Figure 1B. As described in the manuscript, Yoda1 application under these conditions often led to a modest inhibition of the basal current, evident at both hyperpolarized and depolarized potentials. This suggests that Yoda1 may inhibit $I_{Cl,swell}$, likely via a Piezo1-mediated increase in intracellular ionic strength resulting from Na^+ influx, an effect known to suppress VRAC activity.

Consequently, the Yoda1-activated Piezo1 current may be masked by the simultaneous inhibition of Cl^- conductance, making it difficult to resolve under these recording conditions. Furthermore, since Piezo1- and VRAC-mediated currents have similar reversal potentials in our setup, distinguishing between them becomes even more challenging. To more clearly isolate Piezo1 activity, the recordings in Figure 1B were performed under identical ionic conditions but with the addition of DCPIB, in combination with TRAM-34 and paxilline. This allowed effective suppression of $I_{Cl,swell}$ and enabled clearer detection of the Piezo1-mediated current in response to Yoda1.

We hope this response satisfactorily addresses the Reviewer's concern.

4. In the discussion section, the small activation of BK channels is attributed to a possible small increase of intracellular Ca^{2+} through other mechanically gated channels. Can you

indicate what other mechanically gated channels are present in these cells? And if they are known to mediate Ca²⁺ permeation?

We thank the Reviewer for this insightful question. Indeed, other mechanosensitive channels, such as TRPV4, are expressed in GBM cells and are known to be Ca²⁺-permeable. Notably, TRPV4 has been shown to be activated by hypotonic cell swelling and to mediate Ca²⁺ influx in GBM cell lines, including U251 cells (Lanciotti et al., 2012). In light of this, the modest BK channel activation observed in Piezo1 KO cells could plausibly result from Ca²⁺ entry through TRPV4 channels. However, this Ca²⁺ influx appears insufficient to fully activate BK channels at physiologically relevant membrane potentials, and importantly, it does not contribute to volume regulation in U87-MG cells. Supporting this interpretation, a recent study in *Cell Calcium* (Ritzmann et al., 2023) demonstrated that TRPV4-mediated Ca²⁺ influx is not required for hypotonic volume regulation in keratinocytes, which is consistent with our findings.

We have revised the Discussion section of the manuscript to incorporate these points more explicitly (pages 19-20, lines 621-631). *“Indeed, other mechanosensitive channels, such as TRPV4, are expressed in GBM cells and have been reported to mediate Ca²⁺ influx following hypotonic swelling in GBM lines, including U251 cells (Lanciotti et al., 2012). In light of this, the modest BK channel activation observed in Piezo1 KO cells could plausibly result from Ca²⁺ entry through TRPV4 channels. However, this Ca²⁺ influx appears insufficient to effectively activate BK channels at physiologically relevant membrane potentials and, importantly, does not contribute to volume regulation in U87-MG cells. This interpretation is further supported by recent evidence showing that TRPV4-mediated Ca²⁺ influx is dispensable for hypotonic volume regulation in keratinocytes (Ritzmann et al., 2023), consistent with our present findings.”*

5. The current density produced by Piezo1 activation by Yoda1 is small (Figure 1), is this flux of Na⁺ and Ca²⁺ expected to produce the ~20% increase in volume in isotonic conditions (Figure 6)?

We thank the Reviewer for raising this important point. To address whether the small Piezo1 current observed upon Yoda1 activation (~20 pA) could account for the ~20% increase in cell volume under isotonic conditions, we performed a quantitative estimation. Although Piezo1 is permeable to both Na⁺ and Ca²⁺, Na⁺ is the predominant contributor of Yoda1-induced cell swelling. In addition, given that Na⁺ is significantly more concentrated than Ca²⁺ in the extracellular solution (140 vs. 2 mM), the Na⁺ ions transported should be greater than Ca²⁺. Thus, for the purposes of this calculation, we can reasonably treat the Yoda1-induced current as a Na⁺-dominated influx.

Assuming:

- Piezo1 current at -40 mV (resting membrane potential of U87-MG cells) is ~20 pA (this amount is obtained by our previous recordings published by Michelucci et al., 2023 in *Journal of Cellular Physiology*, where at this voltage the Yoda1-mediated Piezo1 current density is on average 0.57 pA/pF divided by the average membrane capacitance of 35 pF)
- Charge per mole of monovalent ion: 96,485 C/mol.

The Na⁺ influx rate (mol/s) is: (20 × 10⁻¹² C/s) / 96,485 C/mol = 2.1 × 10⁻¹⁶ mol/s.

Given that the average cell diameter is $\sim 20 \mu\text{m}$, the corresponding volume is approximately 4.2 pL ($4.2 \times 10^{-12} \text{ L}$). Therefore, the Na^+ concentration increases at a rate of: $2.1 \times 10^{-16} \text{ mol/s} / 4.2 \times 10^{-12} \text{ L} = \approx 50 \mu\text{M/s}$.

After 12 minutes (720 s), corresponding to the peak of the swelling upon Yoda1 application, the cytosolic $[\text{Na}^+]$ would rise by approximately 36 mM. To maintain electroneutrality, we assume a corresponding intracellular retention of Cl^- . This results in a total osmolarity increase of $\sim 72 \text{ mOsm}$ ($36 \text{ mM Na}^+ + 36 \text{ mM Cl}^-$). Given an initial intracellular osmolarity of $\sim 300 \text{ mOsm}$, this represents a 24% increase, which would theoretically lead to a 24% volume increase assuming ideal osmotic behavior ($\Delta V/V_0 \approx \Delta C/C_0$).

This calculation supports the idea that the observed Piezo1 current, although small, is sufficient to account for the measured $\sim 20\%$ cell swelling under isotonic conditions after Yoda1 activation.

We added this calculation to the discussion section, page 22, lines 722-739.

Dear Dr Michelucci,

Re: JP-RP-2025-289215R1 "Piezo1 is an essential player in volume regulation of human glioblastoma cells" by Antonio Michelucci, Maria Vittoria Leonardi, Angela Di Battista, Leonardo Donati, Luigi Sforza, Francesco Morena, Manlio Di Cristina, and Luigi Catacuzzeno

Thank you for submitting your manuscript to The Journal of Physiology. It has been assessed by a Reviewing Editor and by 3 expert referees and we are pleased to tell you that it is acceptable for publication following satisfactory revision.

REVISION CHECKLIST:

We look forward to receiving your revised submission.

Yours sincerely,

Peying Fong
Senior Editor
The Journal of Physiology

REQUIRED ITEMS

- Include a Key Points list in the article itself, before the Abstract.
 - The contact information for the person responsible for 'Research Governance' at your institution needs to be provided (your entry 'N/A' is not permitted). This includes their name and an institutional email address. Please ensure the contact is not an author on this paper and provide an alternate contact if necessary, or confirm in the submission form that the author whose email was provided has sole responsibility for research governance. This is the person who is responsible for regulations, principles and standards of good practice in research carried out at the institution, for instance the ethical treatment of animals, the keeping of proper experimental records or the reporting of results.
 - Your manuscript must include a complete Additional Information section, including competing interests; funding; author contributions and acknowledgements.
 - Papers must comply with the Statistics Policy: https://jp.msubmit.net/cgi-bin/main.plex?form_type=display_requirements#statistics.
- In summary:
- If $n \leq 30$, all data points must be plotted in the figure in a way that reveals their range and distribution. A bar graph with data points overlaid, a box and whisker plot or a violin plot (preferably with data points included) are acceptable formats.
 - If $n > 30$, then the entire raw dataset must be made available either as supporting information, or hosted on a not-for-profit repository, e.g. FigShare, with access details provided in the manuscript.
 - 'n' clearly defined (e.g. x cells from y slices in z animals) in the Methods. Authors should be mindful of pseudoreplication.
 - All relevant 'n' values must be clearly stated in the main text, figures and tables.
 - The most appropriate summary statistic (e.g. mean or median and standard deviation) must be used. Standard Error of the Mean (SEM) alone is not permitted.
 - Exact p values must be stated. Authors must not use 'greater than' or 'less than'. Exact p values must be stated to three significant figures even when 'no statistical significance' is claimed.
 - Please include an Abstract Figure file, as well as the Figure Legend text within the main article file (**we seem to be missing the legend**). The Abstract Figure is a piece of artwork designed to give readers an immediate understanding of the research and should summarise the main conclusions. If possible, the image should be easily 'readable' from left to right or top to bottom. It should show the physiological relevance of the manuscript so readers can assess the importance and

content of its findings. Abstract Figures should not merely recapitulate other figures in the manuscript. Please try to keep the diagram as simple as possible and without superfluous information that may distract from the main conclusion(s). Abstract Figures must be provided by authors no later than the revised manuscript stage and should be uploaded as a separate file during online submission labelled as File Type 'Abstract Figure'. Please also ensure that you include the figure legend in the main article file. All Abstract Figures should be created using BioRender. Authors should use The Journal's premium BioRender account to export high-resolution images. Details on how to use and access the premium account are included as part of this email.

EDITOR COMMENTS

Reviewing Editor:

Thank you for thoroughly addressing the Referees' comments and for submitting a well-revised manuscript. The amendments have improved both the clarity and overall quality of the work.

Please address the further review comments.

Please also see 'Required Items' above.

Senior Editor:

Comments for Authors to ensure the paper complies with the Statistics Policy (Required):
Please refer to the feedback provided by the accompanying report from our Statistics Editor.

Comments to the Author:

Review of your revised manuscript is now complete. You will read in the accompanying critiques that both Expert Referees are satisfied with the revisions incorporated to address scientific aspects they raised during review of the initial submission. Also provided for your reference are comments from our Statistics Editor.

At this time, there remain aspects pertaining to statistical treatment of data reported within the manuscript that require additional attention. Please address the points raised in the Statistics Editor's report. Specifically, these points pertain to the use of mean \pm sem in the time course data, as well as I-V data, that appear in figures 3, 5, 6, and 8, and therefore deviate from the Journal's published policy.

We look forward to receiving your revised manuscript.

REFEREE COMMENTS

Referee #1:

This manuscript presents compelling evidence for a critical role of the mechanosensitive cation/ Ca^{2+} channel Piezo1 in regulating cell volume in the glioblastoma U87-MG cell line.

The findings are likely to attract considerable attention and stimulate further research into the role of Piezo1 in cell volume homeostasis across a range of cell types.

In response to the previous round of critiques, the authors have introduced textual revisions that satisfactorily address all concerns raised by this reviewer.

Referee #2:

The authors have responded to my concerns and comments in this revised version of their manuscript. They have done a great job in addressing my comments and they have improved and clarified their manuscript. This is a fine contribution that enhances our understanding of the roles played by Piezo1 channels and their interplay with other ion channels to regulate important aspects of cell physiology. I have no further comments on this paper.

Referee #3 (statistics review):

The manuscript states that data are presented as mean {plus minus} SD for all figures, except for time course data (Figs. 3A and 8) and average I-V relationships (Figs. 5A-B and 6C-D), which are presented as mean {plus minus} SEM. However, the journal's policy requires that data summaries be presented as mean {plus minus} SD unless the use of SEM is fully justified and accompanied by confidence intervals. No justification for using SEM in these figures is provided, and confidence intervals are not reported alongside the SEM values.

This policy is necessary because the standard deviation (SD) reflects the true variability of the data around the mean, providing a clear picture of the sample's distribution. In contrast, the standard error of the mean (SEM), which primarily indicates the precision of the mean estimate, can obscure the true variability within the data, potentially misleading readers about the spread of the population. Reporting SEM without justification risks underrepresenting data variability and reduces transparency, especially when the SD has been used elsewhere.

Recognising that including confidence intervals may not be practical for these figures, please revise the manuscript to either:

- Present all data as mean {plus minus} SD to comply with the policy, or

- Provide a clear justification for using SEM in Figs. 3A, 5A-B, 6C-D, and 8, explaining why SEM is more appropriate for these specific cases within the Methods section.

END OF COMMENTS

Re: JP-RP-2025-289215R1 "Piezo1 is an essential player in volume regulation of human glioblastoma cells" by Antonio Michelucci, Maria Vittoria Leonardi, Angela Di Battista, Leonardo Donati, Luigi Sforna, Francesco Morena, Manlio Di Cristina, and Luigi Catacuzzeno

Thank you for submitting your manuscript to The Journal of Physiology. It has been assessed by a Reviewing Editor and by 3 expert referees and we are pleased to tell you that it is acceptable for publication following satisfactory revision.

REVISION CHECKLIST:

We look forward to receiving your revised submission.

Yours sincerely,

Peying Fong

REQUIRED ITEMS

- Include a [Key Points](https://jp.msubmit.net/cgi-bin/main.plex?form_type=display_requirements#keypointssummary) list in the article itself, before the Abstract.

Relevant key points have been incorporated in the manuscript to enhance clarity and impact.

- The contact information for the person responsible for 'Research Governance' at your institution needs to be provided (your entry 'N/A' is not permitted). This includes their name and an institutional email address. Please ensure the contact is not an author on this paper and provide an alternate contact if necessary or confirm in the submission form that the author whose email was provided has sole responsibility for research governance. This is the person who is responsible for regulations, principles and standards of good practice in research carried out at the institution, for instance the ethical treatment of animals, the keeping of proper experimental records or the reporting of results.

In response to the request, we have added the contact information (i.e., institutional e-mail) for Prof. Helios Vocca, the person responsible for Research Governance at our institution.

- Your manuscript must include a complete [Additional Information](https://jp.msubmit.net/cgi-bin/main.plex?form_type=display_requirements#addinfo) section, including competing interests; funding; author contributions and acknowledgements.

Added

- Papers must comply with the Statistics Policy: https://jp.msubmit.net/cgi-bin/main.plex?form_type=display_requirements#statistics.

In summary:

- If $n \leq 30$, all data points must be plotted in the figure in a way that reveals their range and distribution. A bar graph with data points overlaid, a box and whisker plot or a violin plot (preferably with data points included) are acceptable formats.

In compliance with the Journal's Statistics Policy have ensured that all individual data points are plotted in the relevant figures in a format that clearly displays their range and distribution.

- If $n > 30$, then the entire raw dataset must be made available either as supporting

information, or hosted on a not-for-profit repository, e.g. FigShare, with access details provided in the manuscript.

We confirm that none of the datasets in our study have $n > 30$. Therefore, the requirement to provide the full raw dataset as supporting information or via an external repository does not apply in our case.

- 'n' clearly defined (e.g. x cells from y slices in z animals) in the Methods. Authors should be mindful of pseudoreplication.

The definition of 'n' is clearly stated in the Methods section.

- All relevant 'n' values must be clearly stated in the main text, figures and tables.

All relevant 'n' values have been explicitly included in the main text, figures, and legends.

- The most appropriate summary statistic (e.g. mean or median and standard deviation) must be used. Standard Error of the Mean (SEM) alone is not permitted.

We have used the most appropriate summary statistics, including mean with standard deviation (SD). We have not used standard error of the mean (SEM).

- Exact p values must be stated. Authors must not use 'greater than' or 'less than'. Exact p values must be stated to three significant figures even when 'no statistical significance' is claimed.

Exact p-values are reported throughout the manuscript, to three significant figures, including where results are not statistically significant.

- Please include an Abstract Figure file, as well as the Figure Legend text within the main article file (**we seem to be missing the legend**). The Abstract Figure is a piece of artwork designed to give readers an immediate understanding of the research and should summarise the main conclusions. If possible, the image should be easily 'readable' from left to right or top to bottom. It should show the physiological relevance of the manuscript so readers can assess the importance and content of its findings. Abstract Figures should not merely recapitulate other figures in the manuscript. Please try to keep the diagram as simple as possible and without superfluous information that may distract from the main conclusion(s). Abstract Figures must be provided by authors no later than the revised manuscript stage and should be uploaded as a separate file during online submission labelled as File Type 'Abstract Figure'. Please also ensure that you include the figure legend in the main article file. All Abstract Figures should be created using BioRender. Authors should use The Journal's premium BioRender account to export high-resolution images. Details on how to use and access the premium account are included as part of this email.

We have created and included an Abstract Figure summarizing the main conclusions and physiological relevance of our study. This figure was prepared using BioRender, as requested, and uploaded as a separate file labeled 'Abstract Figure' during submission. The Figure Legend for the Abstract Figure has been included in the main article file, as required.

EDITOR COMMENTS

Reviewing Editor:

Thank you for thoroughly addressing the Referees' comments and for submitting a well-revised manuscript. The amendments have improved both the clarity and overall quality of the work.

Please address the further review comments.

Please also see 'Required Items' above.

We thank the Reviewing Editor for the positive feedback and are pleased to hear that the revisions have improved the clarity and overall quality of the manuscript.

We carefully addressed the further review comments and ensured that all items listed under 'Required Items' are fully addressed in our next submission.

Thank you again for your continued guidance throughout the review process.

Senior Editor:

Comments for Authors to ensure the paper complies with the Statistics Policy (Required): Please refer to the feedback provided by the accompanying report from our Statistics Editor.

Comments to the Author:

Review of your revised manuscript is now complete. You will read in the accompanying critiques that both Expert Referees are satisfied with the revisions incorporated to address scientific aspects they raised during review of the initial submission. Also provided for your reference are comments from our Statistics Editor.

At this time, there remain aspects pertaining to statistical treatment of data reported within the manuscript that require additional attention. Please address the points raised in the Statistics Editor's report. Specifically, these points pertain to the use of mean \pm sem in the time course data, as well as I-V data, that appear in figures 3, 5, 6, and 8, and therefore deviate from the Journal's published policy.

We look forward to receiving your revised manuscript.

We thank the Senior Editor for the update and are pleased that the scientific revisions have been well received by the Expert Reviewers.

We have carefully reviewed the report from the Statistics Editor and fully acknowledge the remaining concerns regarding the statistical treatment of our data, specifically the use of mean \pm SEM in the time course and I-V data presented in Figures 3, 5, 6, and 8.

In accordance with *The Journal of Physiology's* Statistics Policy, we revised the relevant figures and legends to use the appropriate summary statistics (e.g., mean \pm SD). These changes are clearly indicated in our revised manuscript.

We appreciate the detailed feedback and ensured full compliance with the Journal's statistical guidelines in our next submission.

REFEREE COMMENTS

Referee #1:

This manuscript presents compelling evidence for a critical role of the mechanosensitive cation/ Ca^{2+} channel Piezo1 in regulating cell volume in the glioblastoma U87-MG cell line. The findings are likely to attract considerable attention and stimulate further research into the role of Piezo1 in cell volume homeostasis across a range of cell types.

In response to the previous round of critiques, the authors have introduced textual revisions that satisfactorily address all concerns raised by this reviewer.

We sincerely thank the Reviewer for his/her thoughtful comments. We are grateful for the recognition of the significance of our findings regarding the role of the mechanosensitive channel Piezo1 in regulating cell volume in glioblastoma cells. We also appreciate the acknowledgment that our revisions have satisfactorily addressed Reviewer's previous concerns. The constructive feedback has been invaluable in strengthening our manuscript.

Referee #2:

The authors have responded to my concerns and comments in this revised version of their manuscript. They have done a great job in addressing my comments and they have improved and clarified their manuscript. This is a fine contribution that enhances our understanding of the roles played by Piezo1 channels and their interplay with other ion channels to regulate important aspects of cell physiology. I have no further comments on this paper.

We sincerely thank the Reviewer for his/her kind and supportive comments. We are pleased that our revisions have successfully addressed the Reviewer's concerns and that he/she found the updated manuscript improved and clarified. We are particularly grateful to the Reviewer for recognizing our work as a valuable contribution to advancing the understanding of Piezo1 and its interactions with other ion channels in regulating important mechanisms of cell physiology.

Referee #3 (statistics review):

The manuscript states that data are presented as mean {plus minus} SD for all figures,

except for time course data (Figs. 3A and 8) and average I-V relationships (Figs. 5A-B and 6C-D), which are presented as mean {plus minus} SEM. However, the journal's policy requires that data summaries be presented as mean {plus minus} SD unless the use of SEM is fully justified and accompanied by confidence intervals. No justification for using SEM in these figures is provided, and confidence intervals are not reported alongside the SEM values.

This policy is necessary because the standard deviation (SD) reflects the true variability of the data around the mean, providing a clear picture of the sample's distribution. In contrast, the standard error of the mean (SEM), which primarily indicates the precision of the mean estimate, can obscure the true variability within the data, potentially misleading readers about the spread of the population. Reporting SEM without justification risks underrepresenting data variability and reduces transparency, especially when the SD has been used elsewhere.

Recognising that including confidence intervals may not be practical for these figures, please revise the manuscript to either:

- Present all data as mean {plus minus} SD to comply with the policy, or
- Provide a clear justification for using SEM in Figs. 3A, 5A-B, 6C-D, and 8, explaining why SEM is more appropriate for these specific cases within the Methods section.

In response to the Reviewer's comments, we have revised the manuscript so that all data are now consistently presented as mean \pm SD, including the time course data (Figs. 3A and 8) and the average I-V relationships (Figs. 5A-B and 6C-D). Figure legends and relevant text (i.e., statistical analysis) have been updated accordingly to reflect this change.

We appreciate the Reviewer's guidance in helping us ensure that our data presentation aligns with the journal's statistical reporting standards.

Dear Professor Michelucci,

Re: JP-RP-2025-289215R2 "Piezo1 is an essential player in volume regulation of human glioblastoma cells" by Antonio Michelucci, Maria Vittoria Leonardi, Angela Di Battista, Leonardo Donati, Luigi Sforza, Francesco Morena, Manlio Di Cristina, and Luigi Catacuzzeno

We are pleased to tell you that your paper has been accepted for publication in The Journal of Physiology.

Yours sincerely,

Peying Fong
Senior Editor
The Journal of Physiology

If you would like to receive our 'Research Roundup', a monthly newsletter highlighting the cutting-edge research published in The Physiological Society's family of journals (The Journal of Physiology, Experimental Physiology, Physiological Reports, The Journal of Nutritional Physiology and The Journal of Precision Medicine: Health and Disease), please click this link, fill in your name and email address and select 'Research Roundup':
<https://www.physoc.org/journals-and-media/membernews>

- You can help your research get the attention it deserves! Check out Wiley's free Promotion Guide for best-practice recommendations for promoting your work at: www.wileyauthors.com/eeo/guide. You can learn more about Wiley Editing Services which offers professional video, design, and writing services to create shareable video abstracts, infographics, conference posters, lay summaries, and research news stories for your research at: www.wileyauthors.com/eeo/promotion.

EDITOR COMMENTS

Thank you for incorporating the final changes requested. At this time, your manuscript is ready for final acceptance. Congratulations and thank you for this fine contribution to The Journal of Physiology.